# SMARCA4 loss is synthetic lethal with CDK4/6 inhibition in non-small cell lung cancer

Yibo Xue [1,2], Brian Meehan[3], Zheng Fu[1,2], Xue Qing D. Wang[1], Pierre Olivier Fiset[4], Ralf Rieker[5], Cameron Levins[6], Tim Kong [1,2], Xianbing Zhu[1,2], Geneviève Morin[1,2], Lashanda Skerritt[1,2], Esther Herpel[7], Sriram Venneti[8], Daniel Martinez[9], Alexander R. Judkins[10], Sungmi Jung[4], Sophie Camilleri-Broet[4], Anne V. Gonzalez[11], Marie-Christine Guiot[12], William W. Lockwood[13,14,15], Jonathan D. Spicer[16], Abbas Agaimy[5], William A. Pastor[1,2], Josée Dostie[1], Janusz Rak[3], William D. Foulkes [6,17,18] & Sidong Huang [1,2]

Tumor suppressor *SMARCA4 (BRG1)*, a key SWI/SNF chromatin remodeling gene, is frequently inactivated in cancers and is not directly druggable. We recently uncovered that SMARCA4 loss in an ovarian cancer subtype causes cyclin D1 deficiency leading to susceptibility to CDK4/6 inhibition. Here, we show that this vulnerability is conserved in non-small cell lung cancer (NSCLC), where SMARCA4 loss also results in reduced cyclin D1 expression and selective sensitivity to CDK4/6 inhibitors. In addition, SMARCA2, another SWI/SNF subunit lost in a subset of NSCLCs, also regulates cyclin D1 and drug response when SMARCA4 is absent. Mechanistically, SMARCA4/2 loss reduces cyclin D1 expression by a combination of restricting *CCND1* chromatin accessibility and suppressing c-Jun, a transcription activator of *CCND1*. Furthermore, SMARCA4 loss is synthetic lethal with CDK4/6 inhibition both in vitro and in vivo, suggesting that FDA-approved CDK4/6 inhibitors could be effective to treat this significant subgroup of NSCLCs.

[1] Department of Biochemistry, McGill University, Montreal, QC H3G 1Y6, Canada. [2] The Rosalind & Morris Goodman Cancer Research Centre, McGill University, Montreal, QC H3A 1A3, Canada. [3] Department of Pediatrics, McGill University, and Research Institute of McGill University Health Centre, Montreal Children's Hospital, Montreal, QC H4A 3J1, Canada. [4] Department of Pathology, Glen Site, McGill University Health Centre, Montreal, QC H4A 3J1, Canada. [5] Institute of Pathology, Friedrich-Alexander-University Erlangen-Nürnberg, University Hospital, 91054 Erlangen, Germany. [6] Department of Human Genetics, McGill University, Montreal, QC H3A 0C7, Canada. [7] Tissue Bank of the National Center for Tumor Diseases (NCT) Heidelberg, and Institute of Pathology, Heidelberg University Hospital, 69120 Heidelberg, Germany. [8] Pathology and Neuropathology, University of Michigan Medical School, Ann Arbor, MI 48109-0605, USA. [9] Children's Hospital of Philadelphia Research Institute, Philadelphia, PA 19104, USA. [10] Department of Pathology and Laboratory Medicine, Children's Hospital Los Angeles, Keck School of Medicine of University of Southern California, Los Angeles, CA 90027, USA. [11] Division of Respiratory Medicine, Montreal Chest Institute, McGill University Health Centre, Montreal, QC H4A 3J1, Canada. [12] Department of Pathology, Montreal Neurological Hospital/Institute, McGill University Health Centre, Montreal, QC H4A 3J1, Canada. [13] Department of Integrative Oncology, British Columbia Cancer Agency, Vancouver, BC V5Z 1L3, Canada. [14] Interdisciplinary Oncology Program, University of British Columbia, Vancouver, BC V6T 1Z2, Canada. [15] Department of Pathology and Laboratory Medicine, University of British Columbia, Vancouver, BC V6T 2B5, Canada. [16] Department of Surgery, McGill University Health Center, Montreal, QC H4A 3J1, Canada. [17] Department of Medical Genetics, and Lady Davis Institute, Jewish General Hospital, McGill University, Montreal, QC H3T 1E2, Canada. [18] Department of Medical Genetics and Cancer Research Program, Research Institute of the McGill University Health Centre, McGill University, Montreal, QC H4A 3J1, Canada. Correspondence and requests for materials should be addressed to S.H. (email: sidong.huang@mcgill.ca)

SMARCA4, a catalytic ATPase subunit of SWI/SNF (SWItch/Sucrose Non-Fermentable) chromatin remodeling complexes, is inactivated by mutations or other mechanisms in non-small cell lung cancers (NSCLCs; >10%)[1–8]. Furthermore, concomitant loss of protein expression of SMARCA4 and SMARCA2, another mutually exclusive ATPase subunit of SWI/ SNF, occurs in a NSCLC subset associated with poor prognosis[2]. While approximately 20% of SMARCA4 mutations in NSCLCs co-occur with *KRAS* mutations, the remaining cases rarely co-occur with known druggable oncogenic mutations[1,3,4,8]. Previous studies have uncovered several synthetic lethal interactions of SMARCA4 loss in NSCLC cells, including suppression of SMARCA2[8,9], a non-catalytic activity of EZH2[10], and aurora kinase A[11]. However, these vulnerabilities associated with SMARCA4 deficiency are currently not druggable with any FDA-approved agents. Thus, SMARCA4-deficient NSCLCs still lack an effective targeted treatment option.

In addition to NSCLC, inactivating *SMARCA4* mutations are known to be the sole genetic driver event in ~100% of small cell carcinoma of the ovary, hypercalcemic type (SCCOHT)[12–14], which, unlike NSCLC, has a remarkably simple genome that harbors few mutations or chromosomal alterations[15,16]. Using kinome-focused RNA interference (RNAi) screens, we recently uncovered that SCCOHT cells are selectively sensitive to cyclin-dependent kinase 4/6 (CDK4/6) inhibition[17]. We found that SMARCA4 loss causes profound downregulation of cyclin D1, which limits CDK4/6 kinase activity in SCCOHT cells and results in less buffering against CDK4/6 inhibition. Our unexpected findings thus extend the initial application of CDK4/6 inhibitors in treating estrogen receptor-positive (ER+) breast cancers which are often characterized with dysregulated CDK4/6 activation[18–25], where the oncogenic addiction to cyclin D1 is being targeted. In the case of SCCOHT, the critically low level of cyclin D1 caused by SMARCA4 loss is a cancer vulnerability that can also be targeted by the same inhibitors.

Here, we investigated this synthetic lethal interaction in SMARCA4-deficient NSCLC, which has a complex mutation landscape, and explored the potential strategy of using CDK4/6 inhibitors to treat this highly aggressive subgroup of lung cancer.

## Results

**Reduced cyclin D1 in SMARCA4-deficient NSCLC causes sensitivities to CDK4/6 inhibitors**. We first examined the potential correlation between SMARCA4 status and cyclin D1 expression in NSCLC cells as seen in SCCOHT. Despite the differences in tissue origins and mutation burdens between these two cancer types, SMARCA4-deficient NSCLC cell lines ($n = 11$) also express lower levels of cyclin D1 protein (Fig. 1a) and messenger RNA (mRNA; Fig. 1b) compared to SMARCA4-proficient NSCLC cell lines ($n = 9$), which consist of the major NSCLC subtypes with *KRAS* ($n = 4$) or *EGFR* ($n = 3$) mutations as well as those without these driver mutations ($n = 2$; Supplementary Data 1). Among the 4 SMARCA4/2-dual deficient cell-lines, H1703 and H522 do not harbor *KRAS* mutations and express the lowest levels of cyclin D1 protein and mRNA (Fig. 1a, b), suggesting that SMARCA2 may also regulate cyclin D1 expression. Such correlation was not observed for key cell cycle regulators of G1–S-phase transition (Supplementary Fig. 1). In addition, almost all SMARCA4-deficient cell lines retain retinoblastoma (RB) and are negative or express lower levels of the CDK4/6 inhibitor p16[INK4a] (Fig. 1a), a profile known to be associated with positive responses to CDK4/6 inhibitors[20–23].

Since SMARCA4-deficient SCCOHT cells are vulnerable to CDK4/6 inhibition attributed to cyclin D1 deficiency[17], we investigated this susceptibility in above mentioned lung cancer cell lines. Indeed, SMARCA4-deficient NSCLC cells are also highly sensitive to the CDK4/6 inhibitor palbociclib in both cell viability (Fig. 1c) and long-term colony formation assays (Fig. 1d and Supplementary Fig. 2a), contrasting to SMARCA4-proficient/ *KRAS* wild-type (WT) cells. *KRAS* mutants served as a positive control-*KRAS* mutations in NSCLC are known to be synthetic lethal with CDK4 inhibition[26] and the CDK inhibitor abemaciclib has shown single-agent antitumor activity in patients with *KRAS*-mutant NSCLCs[27]. We found that SMARCA4-deficient cell lines, regardless of the *KRAS* status, have similar palbociclib sensitivities as *KRAS* mutants (Fig. 1c, d and Supplementary Fig. 2a).

Similar results were obtained using abemaciclib (Supplementary Fig. 3a). This differential drug sensitivities of these cell lines do not correlate with their proliferation rates (Supplementary Fig. 4). Consistent with the growth response, palbociclib and abemaciclib suppressed phosphorylation of RB, a direct target of CDK4/6, in SMARCA4-deficient and *KRAS*-mutant cells but not the SMARCA4-proficient/*KRAS* WT cells (Supplementary Fig. 2b, 3b). Furthermore, palbociclib treatment in SMARCA4-deficient NSCLC cells induces strong G1 cell cycle arrest (Fig. 1e, f) but not cell death as indicated by the lack of Annexin V staining (Supplementary Fig. 5). Together, these data show that CDK4/6 inhibitors are effective in inhibiting proliferation of SMARCA4-deficient NSCLC cells predominantly through cell cycle suppression.

We noted that two of the four SMARCA4/2-dual deficient cell lines, H1703 and A427, are among the most sensitive to CDK4/6 inhibitors (Fig. 1c, d, Supplementary Fig. 2a, 3a). The other two SMARCA4/2-dual deficient cell lines, H23 and H522, are both RB deficient and are relatively resistant to CDK4/6 inhibitors (Fig. 1a, c, Supplementary Fig. 2a, 3a). This is likely due to the requirement of RB for palbociclib response[20–23]. Consistent with this, RB knockdown in SMARCA4/2-dual deficient H1703 cells confers strong drug resistance to palbociclib (Supplementary Fig. 6). Thus, these observations suggest that SMARCA4/2-dual deficient cells are more sensitive to CDK4/6 inhibition, if RB expression is intact.

Supporting the role of cyclin D1 deficiency mediating drug sensitivity to CDK4/6 inhibition, ectopic cyclin D1 expression confers palbociclib resistance in both SMARCA4-deficient H1299 and SMARCA4/2-dual deficient H1703 (Fig. 1g, h). Complementarily, cyclin D1 knockdown in SMARCA4/2-proficient HCC827 and PC9 controls enhances drug responses (Fig. 1i, j). Collectively, these data demonstrate that SMARCA4 deficiency is associated with reduced cyclin D1 expression and susceptibility to CDK4/6 inhibition in NSCLC, which is in line with our findings in SCCOHT.

**Palbociclib suppresses SMARCA4-deficient NSCLC in vivo**. To validate our above cell line findings in vivo, we examined the responses of SMARCA4-deficient NSCLC tumors to palbociclib using mouse xenograft models. Even though *KRAS* status does not impact drug sensitivities of SMARCA4-deficient cells in vitro (Fig. 1c, d and Supplementary Fig. 2, 3), we tested two SMARCA4-deficient/*KRAS* WT models to rule out potential contributions of *KRAS* mutations to drug responses in vivo: H1299 (SMARCA4-deficient) and H1703 (SMARCA4/2-dual deficient). As shown in Fig. 2a–d, palbociclib treatment elicited a potent growth inhibition of both tumor models. Immunohistochemistry (IHC) analysis of tumors at the treatment endpoints showed that RB phosphorylation, Ki67 expression and mitotic index were significantly suppressed in palbociclib-treated cohorts (Fig. 2e–h), confirming the target modulation by palbociclib. These results demonstrate that palbociclib is effective in treating SMARCA4-deficient NSCLC tumors in vivo.

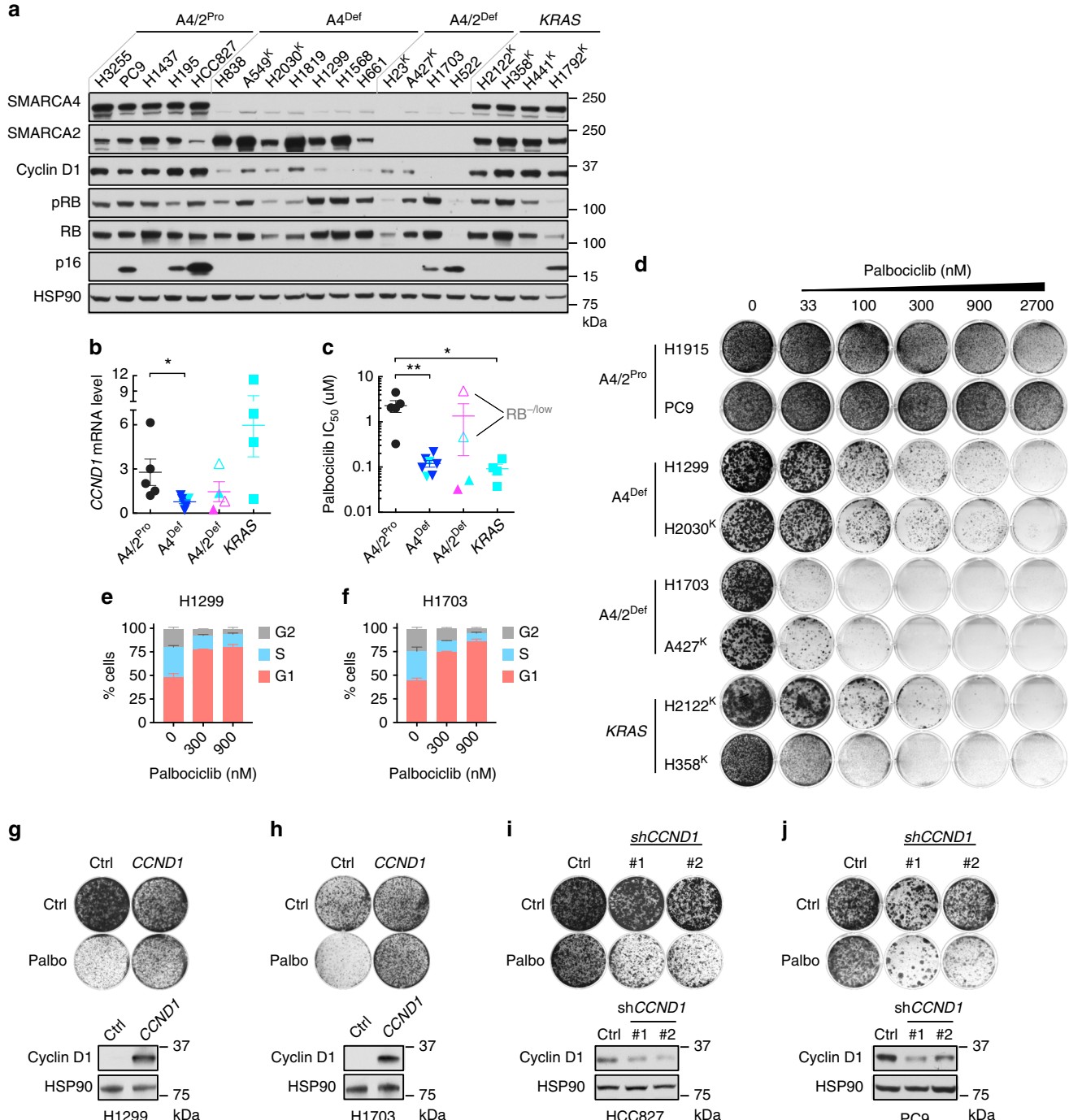

**Fig. 1** Reduced cyclin D1 in SMARCA4-deficient non-small cell lung cancer (NSCLC) cells causessensitivities to cyclin-dependent kinase 4/6 (CDK4/6) inhibitors. **a**, **b** SMARCA4-deficient NSCLC cell lines express reduced cyclin D1 levels. Western blot analysis for the indicated proteins (**a**) and *CCND1* messenger RNA (mRNA) expression (**b**) of a panel of NSCLC cell lines. HSP90 was used as a loading control. Relative *CCND1* mRNA expression (relative to *GAPDH*) was measured by real-time quantitative reverse transcription PCR (RT-qPCR). A4: SMARCA4, A4/2: SMARCA4/2, Pro: proficient, Def: deficient, K: *KRAS* mutation. Empty triangles indicate RB-deficient cell lines. Turquoise color indicates cell lines with *KRAS* mutation. Error bars: mean ± standard deviation (s.d.) of biological replicates ($n = 3$); two-tailed *t* test, *$p < 0.05$. **c**, **d** SMARCA4-deficient NSCLC cells are highly sensitive to palbociclib treatment, similar to *KRAS* mutation cells. **c** Half-maximal inhibitory concentration ($IC_{50}$) of palbociclib in the above cell line panel was determined by measuring cell viability using CellTiter-Blue assay. Error bars: mean ± s.d. of biological replicates ($n = 4$); two-tailed *t* test, *$p < 0.05$, **$p < 0.01$. **d** Colony formation assays of the representative cell lines. Cells were cultured in the absence or presence of palbociclib at the indicated concentrations for 10–14 days. For each cell line, all dishes were fixed at the same time. **e**, **f** Palbociclib treatment in SMARCA4-deficient NSCLC cells induces strong G1 cell cycle arrest. H1299 (**e**) and H1703 (**f**) cells treated with palbociclib for 24 h were fixed, stained with propidium iodide and analyzed by flow cytometry using the Guava easyCyte HT System. **g**, **h** Ectopic expression of cyclin D1 confers drug resistance to palbociclib in H1299 (**g**) and H1703 (**h**) cells. Upper, colony formation assays; lower, immunoblot of cells with stable ectopic expression of *GFP* or *CCND1* and treated with palbociclib (H1299, 300 nM; H1703, 33 nM). **i**, **j** Cyclin D1 knockdown sensitizes HCC827 (**i**) and PC9 (**j**) cells to palbociclib. Upper, colony formation assays in the absence or presence of 300 nM palbociclib; lower, immunoblot of cells expressing pLKO control or short hairpin RNAs (shRNAs) targeting *CCND1*

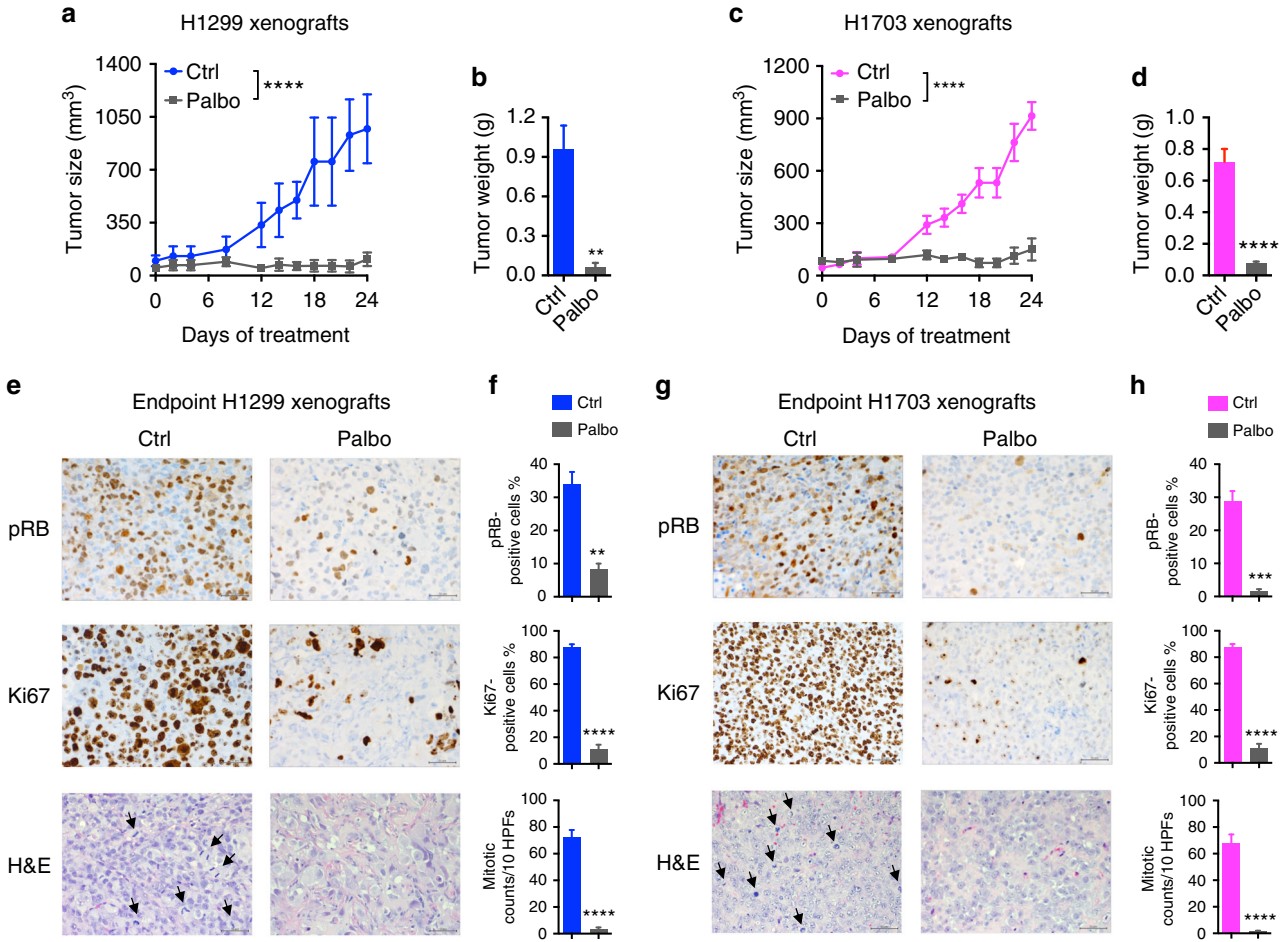

**Fig. 2** Palbociclib is effective against SMARCA4-deficient non-small cell lung cancer (NSCLC) tumor growth in vivo. Palbociclib inhibits tumor growth in xenograft models of H1299 (**a**, **b**, **e**, **f**) and H1703 (**c**, **d**, **g**, **h**). **a**, **c** Tumor size from day 0 of treatment in H1299 (**a**, $n = 4$ per group) and H1703 (**c**, $n = 8$ for vehicle, $n = 7$ for palbociclib; 150 mg kg$^{-1}$) models. Error bars represent mean ± standard error of mean (s.e.m.); two-way analysis of variance (ANOVA), ****$p < 0.0001$. **b**, **d** Final tumor weight measured after surgery in H1299 (**b**) and H1703 (**d**) models. Two-tailed $t$-test, **$p < 0.01$, ****$p < 0.0001$. **e–h** Palbociclib treatment resulted in suppression of RB phosphorylation, Ki67 expression and mitotic index in xenograft tumors of the trial endpoints. Representative images of Immunohistochemistry (IHC) (p-RB, Ki67) and hematoxylin and eosin (H&E) analysis of H1299 (**e**) and H1703 (**g**) xenograft tumor tissues. Bar 50 μm; black arrows point to mitotic active cells as examples. **f**, **h** Quantifications of p-RB, Ki67 and mitotic count of H1299 (**f**, $n = 3$) and H1703 (**h**, $n = 4$). Two-tailed $t$-test, **$p < 0.01$, ***$p < 0.001$, ****$p < 0.0001$

**SMARCA4/2 loss causes reduced cyclin D1 expression in NSCLC.** The correlation between SMARCA4 status and cyclin D1 levels in the cell line panel suggests that SMARCA4 loss causes reduced cyclin D1 expression in NSCLC. Supporting this, restoration of SMARCA4 elevates cyclin D1 expression in SMARCA4-deficient cell lines (H1299, H1703, H2030, A427) regardless of *KRAS* mutation status (Fig. 3a, b and Supplementary Fig. 7a, b). Our cell line panel analysis (Fig. 1a–d and Supplementary Fig. 1–3) also suggests that SMARCA2 plays a role in regulating cyclin D1 expression and drug response to CDK4/6 inhibitors in SMARCA4-deficient NSCLC cells. In line with this, SMARCA2 knockdown in SMARCA4-deficient H1299 cells suppresses cyclin D1 expression (Fig. 3c); conversely, SMARCA2 restoration in SMARCA4/2-dual deficient cells (H1703 and A427) upregulates cyclin D1 protein and mRNA expression (Fig. 3d and Supplementary Fig. 7c). In contrast, SMARCA2 knockdown in SMARCA4/2-proficient H1915 cells has no effect on cyclin D1 expression while knockdown of SMARCA4 in the same cells strongly suppresses cyclin D1 (Supplementary Fig. 8). These data suggest a model in which SMARCA4 plays a dominant role in regulating cyclin D1 expression and drug response to

CDK4/6 inhibitors in NSCLC cells, but SMARCA2 becomes limiting when SMARCA4 is absent.

In keeping with our in vitro findings, we observed a mild but significant correlation ($r = 0.33$; $p = 0.002$) between *SMARCA4* and *CCND1* mRNA expression in a cohort of 83 lung adenocarcinomas (LUADs), a main NSCLC subtype (Fig. 3e). Similar correlation ($r = 0.36$; $p < 0.0001$) was also obtained in a second cohort of 230 LUAD patient tumor samples from The Cancer Genome Atlas (TCGA)[4] (Fig. 3f). Furthermore, we found a strong correlation ($r = 0.8$; $p = 0.001$) between *SMARCA2* and *CCND1* mRNA expression in *SMARCA4*-mutated LUAD tumors (Fig. 3g) but not in the non-selected TCGA cohort (Supplementary Fig. 9), supporting the role of SMARCA2 in regulating *CCND1* expression when SMARCA4 is lost in NSCLC. Using IHC, we next analyzed protein expression of cyclin D1 in a third cohort of NSCLC patient tumors ($n = 93$). In line with tumor mRNA analysis, SMARCA4 IHC-negative NSCLC tumors ($n = 11$) expressed lower levels of cyclin D1 compared to SMARCA4 IHC-positive NSCLCs ($n = 82$) although not statistically significant ($p = 0.105$; Fig. 3h, i). A majority (8/11) of the SMARCA4 IHC-negative tumors

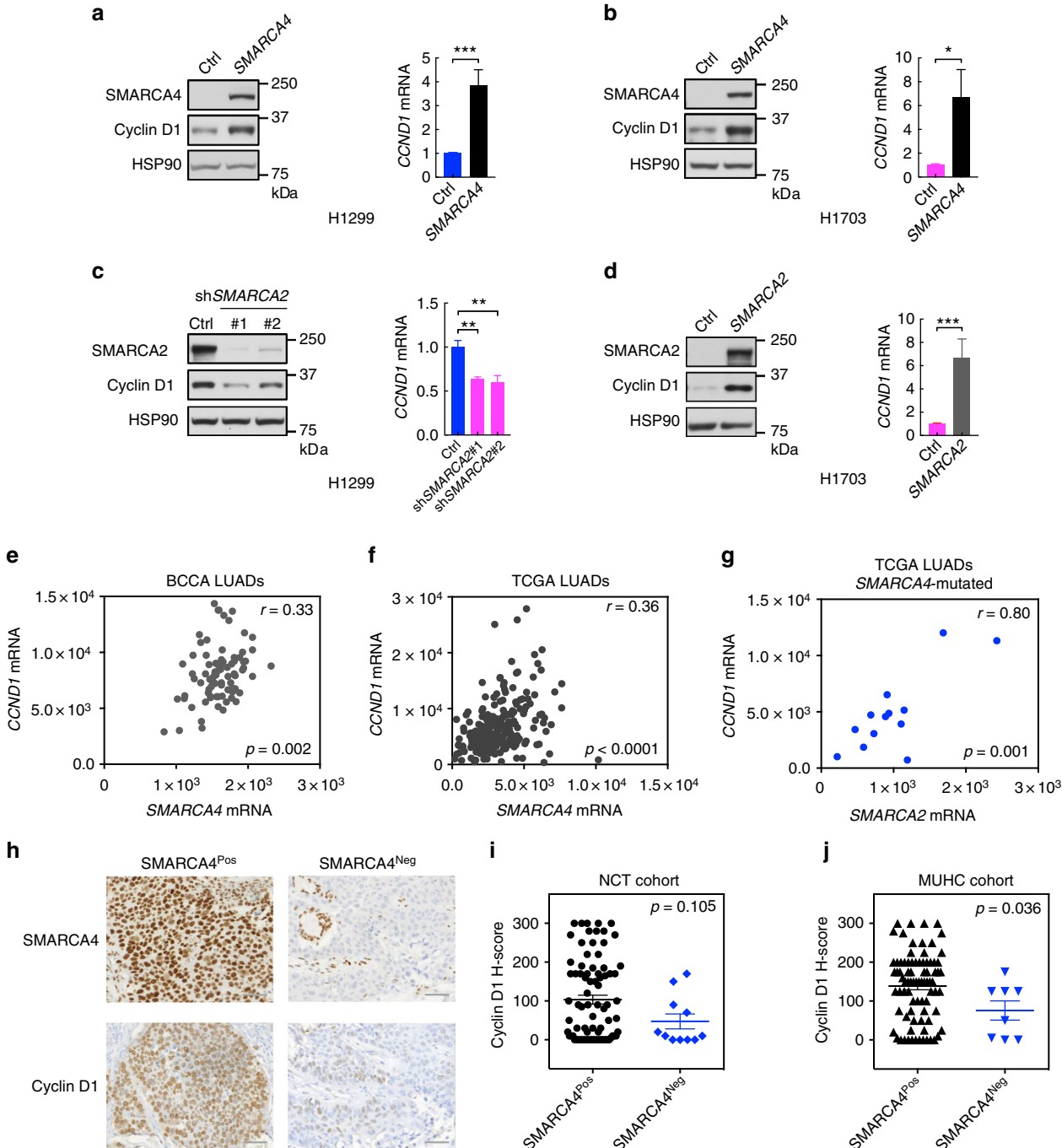

**Fig. 3** SMARCA4/2 loss causes reduced cyclin D1 expression in non-small cell lung cancer (NSCLC). **a–d** SMARCA4/2 regulate cyclin D1 expression in NSCLC. **a, b** SMARCA4 restoration upregulates cyclin D1 protein (left) and messenger RNA (mRNA) (right) expression in H1299 (**a**) and H1703 (**b**) cells. **c** SMARCA2 knockdown in H1299 cells suppresses cyclin D1 protein (left) and mRNA (right) expression. **d** SMARCA2 restoration in H1703 cells elevates cyclin D1 protein (left) and mRNA (right) expression. Relative *CCND1* mRNA expression (relative to *GAPDH*) was measured by real-time quantitative reverse transcription PCR (RT-qPCR). Error bars: mean ± s.d. of biological replicates ($n = 3$, two-tailed $t$-test, $*p < 0.05$, $**p < 0.01$, $***p < 0.001$). **e, f** Correlation of *CCND1* and *SMARCA4* mRNA expression in two cohorts of lung adenocarcinomas (LUADs) from BC Cancer Agency (BCCA; $n = 83$, **e**) and The Cancer Genome Atlas (TCGA; $n = 230$, **f**). **g** Correlation of *CCND1* and *SMARCA2* mRNA expression in SMARCA4-mutated LUADs ($n = 13$) in the TCGA cohort. $r$, Pearson's correlation coefficient. **h–j** Immunohistochemistry (IHC) analysis of cyclin D1 protein expression in two cohorts of NSCLC patient tumors: NCT ($n = 93$; **h, i**) and McGill University Health Center (MUHC; $n = 91$; **j**). Representative IHC images of SMARCA4 IHC-negative tumors are shown; a SMARCA4/2 IHC-positive tumor served as staining control (**h**). Cyclin D1 in IHC analysis was quantified with H-score and analyzed by Wilcoxon rank sum test (**i, j**)

retained RB expression and were p16 negative (Supplementary data 2), a profile associated with positive responses to palbociclib[20–23]. We further confirmed the reduced cyclin D1 protein expression in SMARCA4 IHC-negative NSCLCs in another cohort of 100 patient tumors ($p = 0.036$; Fig. 3j). Collectively, these results support our in vitro data and indicate that SMARCA4-deficient NSCLC tumors express reduced cyclin D1 and may respond to CDK4/6 inhibitors.

**SMARCA4 loss restricts *CCND1* chromatin accessibility**. We next investigated the mechanism by which SMARCA4/2 regulate cyclin D1 expression in NSCLC. Given the chromatin remodeling role of SWI/SNF, we performed assay for transposase-accessible chromatin sequencing (ATAC-seq) in H1703 cells before and after restoration of SMARCA4 or SMARCA2 to examine their global effects on chromatin accessibility in NSCLC cells (Supplementary Data 3, 4). Restoration of SMARCA4 resulted in a dramatic opening of the chromatin landscape: the total number of open chromatin sites more than doubled (Fig. 4a), with the overwhelming majority of new open chromatin sites present at distal, non-promoter sites (Fig. 4b). This observation is consistent with the widespread finding that enhancer utilization varies more between cell types than promoter openness[28,29] and that SMARCA4 opens enhancers in other contexts[30–33]. Expression of SMARCA2 in H1703 cells had an extremely similar global effect to expression of SMARCA4, with SMARCA4-dependent open chromatin sites showing similar magnitude of opening upon SMARCA2 expression (Fig. 4c, d). Using the same isogenic cell pairs, we also performed chromatin immunoprecipitation sequencing (ChIP-Seq) experiments of H3K27Ac, a mark of active promoters and enhancers, and observed gain of H3K27Ac in the vicinity of SMARCA4/2-dependent open chromatin (Fig. 4e, f). Regions of conserved openness in the control and SMARCA4-restored H1703 cells showed similar H3K27Ac in the three conditions (Supplementary Fig. 10). Taken together, these data indicate that a large number of regulatory elements are activated by SMARCA4/2.

Next, we focused on the chromatin architecture of the *CCND1* locus and the potential regulation by SMARCA4/2 (Fig. 5a). Using published ChIP-seq data of H1299 cells expressing inducible SMARCA4[34], we observed that SMARCA4 is present at the *CCND1* promoter. We also confirmed this SMARCA4 occupancy in SMARCA4-restored H1703 cells using ChIP-seq (Fig. 5a) and in the SMARCA4-proficient H1915 cells by ChIP-PCR (Supplementary Fig. 11a). Similarly, SMARCA2 binding was also enriched at the same *CCND1* promoter region in H1299 cells (Supplementary Fig. 11b). These data suggest that SMARCA4/2 may directly regulate *CCND1* expression. Consistent with this, H3K27Ac signal at the *CCND1* promoter region and ATAC-seq signal in the *CCND1* gene body were both elevated upon SMARCA4/2 restoration (Fig. 5a), indicating enhanced chromatin accessibility at the *CCND1* locus when SMARCA4/2 are present. Supporting this, we detected a significant enrichment of E2F1 at the promoter region upon SMARCA4 restoration in H1703 cells (Supplementary Fig. 12a), while E2F1 total expression was not changed (Supplementary Fig. 12b). Given that E2F1 is known to suppress *CCND1*[35], this enhanced E2F1 occupancy may not explain the *CCND1* induction by SMARCA4. Consistent with this, we observed that E2F1 knockdown results in slight upregulation of cyclin D1 (Supplementary Fig. 12b). Nevertheless, enhanced E2F1 binding supports the increased chromatin accessibility of the *CCND1* locus induced by SMARCA4/2, which likely also promotes binding of other transcription factors known to directly activate *CCND1*[36].

In addition to the enhanced accessibility at the *CCND1* promoter, we also observed the strong opening of a SMARCA4-dependent putative enhancer site 50 kb upstream of *CCND1*, which is the only annotated gene locus within the vicinity (Fig. 5a). This region contained two strong-ATAC-seq peaks, with the summit of each peak containing a canonical adaptor protein-1 (AP-1) site, the motif bound by c-Fos/c-Jun dimers[37,38] (Fig. 5b). While c-Fos/c-Jun are known activators of *CCND1*[39,40], their involvement in this putative enhancer is not known. Consistent with a potential enhancer nature, ChIP-Seq signals of H3K27Ac but not H3K4me3, a promoter activation mark, was

significantly increased at these two peaks in response to SMARCA4/2 restoration (Fig. 5a). Analysis of the publicly available ChIP-seq data sets in human umbilical vein endothelial cell (HUVEC; GSM935585, GSM935278) indicates a strong enrichment for c-Fos and c-Jun at these two peaks (Fig. 5b). Together, these data suggest that c-Fos/c-Jun may be involved in SMARCA4/2-mediated *CCND1* regulation through this putative enhancer.

Interestingly, we identified SMARCA4 occupancy at the *JUN* promoter as well as extensive chromatin opening of the *JUN* locus upon SMARCA4/2 restoration in H1703 cells (Fig. 5c). In line with this observation, restoration of SMARCA4 upregulates c-Jun mRNA and protein expression in H1703 cells (Fig. 5d, e). This effect was also observed to a lesser extent in H1299 (Fig. 5f, g), as these cells are proficient in SMARCA2. Consistent with this, SMARCA2 restoration in H1703 cells also upregulates c-Jun expression (Fig. 5h, i). Supporting the regulation of *JUN* by SMARCA4/2, we observed a trend of lower *JUN* mRNA expression in LUAD patient tumors expressing low *SMARCA4/2* (Supplementary Fig. 13). In contrast, we found that c-Fos is not regulated by SMARCA4/2 (Supplementary Fig. 14). Importantly, knockdown of c-Jun or c-Fos partially abrogated SMARCA4-mediated induction of cyclin D1 mRNA and protein expression (Fig. 5j, k and Supplementary Fig. 14). Collectively, these data suggest that, in addition to direct modulating chromatin accessibility of the *CCND1* locus, SMARCA4/2 also promote cyclin D1 expression by inducing c-Jun (Fig. 5l).

**SMARCA4 loss is synthetic lethal with CDK4/6 inhibition**. Finally, we sought to further establish the synthetic lethal interaction between SMARCA4 loss and CDK4/6 inhibition. In contrast to SCCOHT, SMARCA4-deficient NSCLC cells (H1299, H2030, H1703, A427) can better tolerate restoration of SMARCA4 expression, which results in drug resistance to CDK4/6 inhibition, regardless of *KRAS* mutation status (Fig. 6a, b and Supplementary Fig. 7d, e). In keeping with the role of SMARCA2 in regulating drug responses, SMARCA2 knockdown in SMARCA4-deficient H1299 cells sensitizes cells to palbociclib treatment (Fig. 6c); conversely, SMARCA2 restoration in SMARCA4/2-dual deficient cells (H1703 and A427) confers resistance to palbociclib (Fig. 6d and Supplementary Fig. 7f). Thus, these in vitro data support the synthetic lethal interaction between SMARCA4 loss and CDK4/6 inhibition in NSCLC.

To further verify this in vivo, we generated mouse xenograft models using an isogenic cell pair of H1299 engineered to only differ in SMARCA4 status. H1299 was chosen as this cell line can better tolerate forced expression of SMARCA4 (Fig. 6a). Consistent with our in vitro data, SMARCA4 restoration does not significantly suppress H1299 tumor grow but leads to drug resistance to palbociclib treatment compared to the control tumors (Fig. 6e–g). Together, our data demonstrate that SMARCA4 loss in NSCLC results in reduced cyclin D1 expression, which underlies the synthetic lethal interaction between SMARCA4 deficiency and CDK4/6 inhibition.

## Discussion
Our findings show that the synthetic lethal interaction between SMARCA4 loss and CDK4/6 inhibition, mediated by cyclin D1 deficiency, is a common druggable vulnerability of NSCLC and SCCOHT, despite their differences in tissue of origin and mutation landscape. In this study, we also establish the role of SMARCA2 in regulating cyclin D1 and drug response to CDK4/6 inhibitors. Furthermore, we provide mechanistic insights for the regulation of cyclin D1 by SMARCA4/2 in NSCLC, where

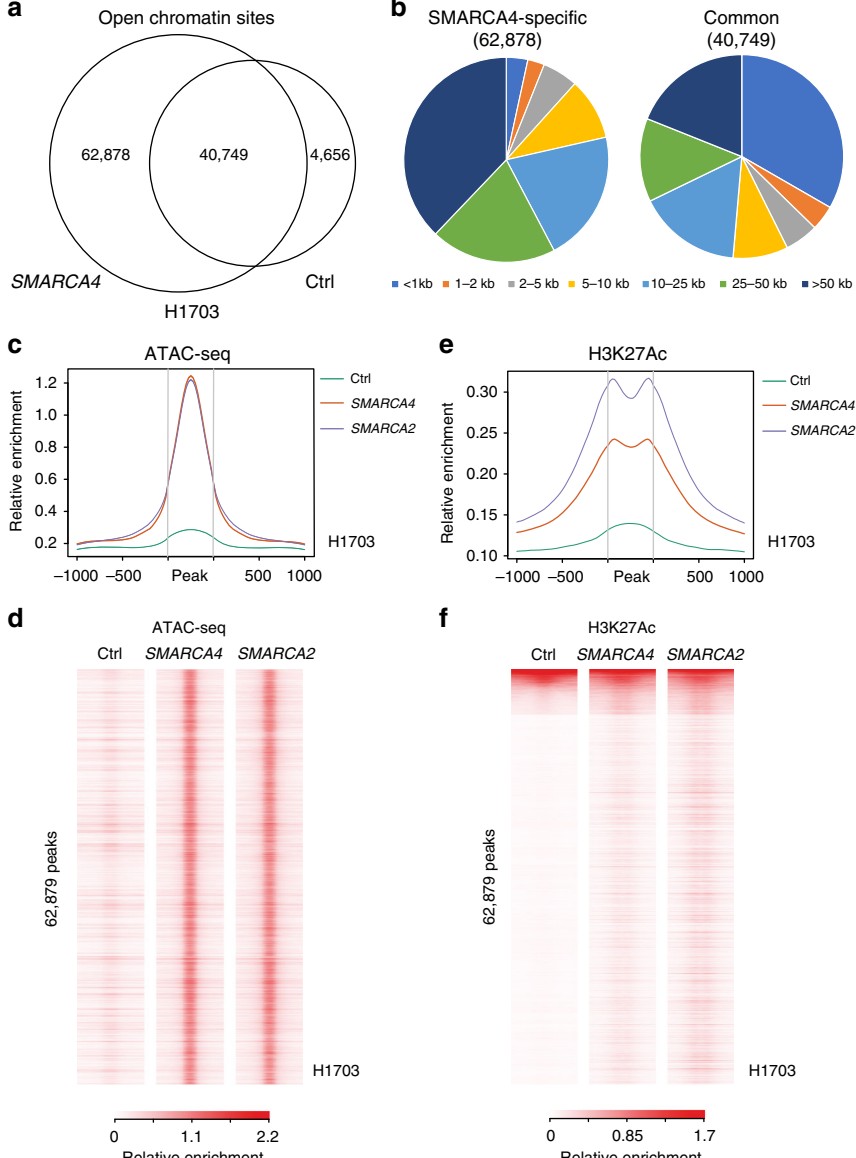

**Fig. 4** Extensive opening of regulatory elements by induction of SMARCA4/2. **a** Venn diagram showing overlap of open chromatin sites in control infected H1703 cells with or without SMARCA4 overexpression. Note the dramatic increase in open chromatin sites upon SMARCA4 overexpression. **b** Distribution of SMARCA4-dependent and -independent open chromatin sites relative to nearest gene transcriptional start site. Note that SMARCA4-dependent sites are much less likely to be promoters (within 1 kb of transcription start site (TSS)). **c**, **d** Metaplot (**c**) and heatmap (**d**) of assay for transposase-accessible chromatin sequencing (ATAC-seq) read data from control-infected, SMARCA4-infected and SMARCA2-infected cells over the 62,878 SMARCA4-dependent ATAC peaks. Note similar effect of SMARCA2 and SMARCA4. **e**, **f** Metaplot (**e**) and heatmap (**f**) of H3K27Ac chromatin immunoprecipitation (ChIP) data from control-transfected, SMARCA4-infected and SMARCA2-infected cells over the 62,878 SMARCA4-dependent ATAC peaks

SMARCA4/2 remodel the chromatin structure of the *CCND1* locus and its transcription activator *JUN*.

Our ATAC-seq and H3K27Ac ChIP-seq results support that SMARCA4/2 positively regulate cyclin D1 expression by enhancing chromatin accessibility at the *CCND1* locus. This chromatin opening may promote accessibility to transcription factors that directly regulate *CCND1*[36], which requires further investigation. Our genomic studies also identify a putative enhancer 50 kb upstream of *CCND1* which contains canonical AP-1 site motif bound by c-Fos/c-Jun dimers and is clearly opened by SMARCA4/2. Indeed, we found evidence of c-Fos/c-Jun occupancy at this site in a publicly available ChIP-seq data set. While the relevance of this putative enhancer site remains to be

established in future studies, we found that SMARCA4-induced *CCND1* upregulation partially requires c-Fos/c-Jun. Furthermore, SMARCA4/2 activate c-Jun expression potentially through directly regulating the chromatin structure of the *JUN* locus, as observed in our chromatin accessibility assays and ChIP experiments as well as in publicly available data tracks. Thus, our data suggest a model in which SMARCA4/2 regulates cyclin D1 by a combination of direct activation of the *CCND1* promoter and by positive regulation of *JUN* which also activates *CCND1*[39,40].

Our data do not rule out additional mechanisms by which SMARCA4/2 regulates cyclin D1 expression. It is also possible that cyclin D1 deficiency caused by SMARCA4 loss may be compensated by other regulators of cell cycle progression. For

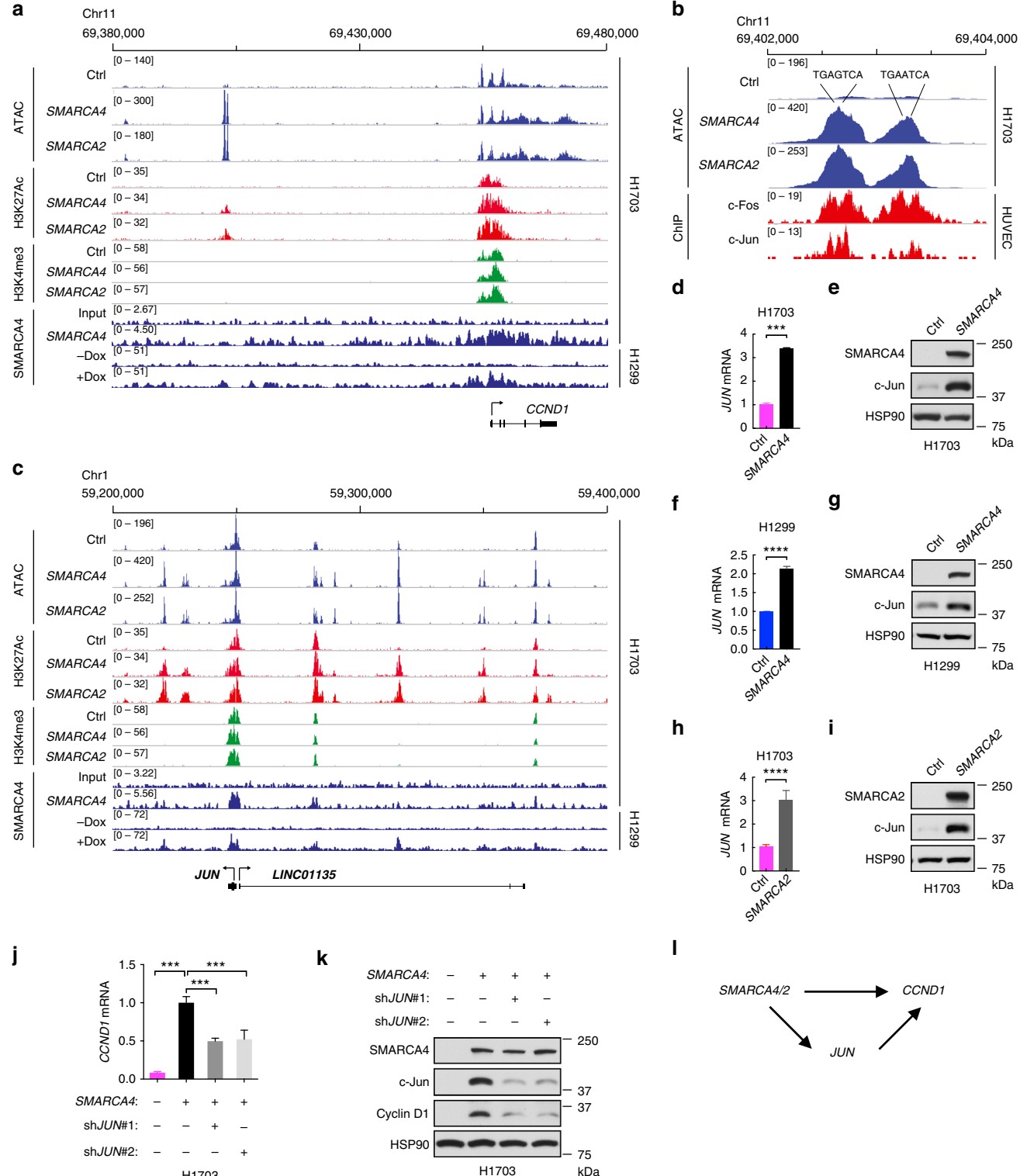

example, we found elevated CDK2 expression in SMARCA4-deficient NSCLC cells which may help to maintain RB phosphorylation. In addition, other potential dysregulations of cell cycle progression as well as the oncogenic driver pathways caused by SMARCA4 loss remain to be investigated in NSCLC.

In summary, our study demonstrates that the synthetic lethal interaction between SMARCA4 loss and CDK4/6 inhibition is a vulnerability of NSCLC that can be exploited therapeutically. Abemaciclib has shown single-agent antitumor activity in patients with *KRAS*-mutant NSCLCs[27] and palbociclib is being investigated in clinical trials (NCT02022982, NCT03170206, NCT02152631). Thus, our study suggests that SMARCA4-deficient NSCLC patients, a significant subgroup of this aggressive disease, may also benefit from this treatment strategy using CDK4/6 inhibitors. Given that SMARCA4 is also inactivated in a variety of other cancer types[6], this common druggable vulnerability shared by SCCOHT and NSCLC may also be conserved in other SMARCA4-deficient tumors.

**Fig. 5** SMARCA4/2 regulate *CCND1* via controlling chromatin accessibility and upregulating *JUN*. **a** Assay for transposase-accessible chromatin sequencing (ATAC-seq) and chromatin immunoprecipitation sequencing (ChIP-seq) data in vicinity of the *CCND1* locus indicate enhanced chromatin accessibility upon SMARCA4/2 restoration. Note SMARCA4 at *CCND1* promoter and formation of new putative enhancer ~50 kb upstream of *CCND1* promoter. All data were generated in H1703 cells before and after restoration of SMARCA4 or SMARCA2, except the publicly available SMARCA4 ChIP data in H1299 cells expressing doxycycline (Dox)-inducible SMARCA4[34]. Track height is normalized to relative number of mapped reads. **b** Zoomed-in view of the putative *CCND1* enhancer region. Shown are ATAC-seq peaks in H1703 cells before and after SMARCA4/2 restoration and the publicly available c-Fos/c-Jun ChIP data of endothelial cell line, human umbilical vein endothelial cell (HUVEC) (GSM935585, GSM935278). Location of canonical adaptor protein-1 (AP-1) motifs are indicated. **c** ATAC and ChIP-seq data in vicinity of *JUN* locus as described in **a**. Note SMARCA4 at *JUN* promoter and extensive opening of nearby putative enhancers. **d–i** Restoration of SMARCA4 in H1703 (**d**, **e**) and H1299 (**f**, **g**) or SMARCA2 restoration in H1703 (**h**, **i**) cells upregulate c-Jun messenger RNA (mRNA) (**d**, **f**, **h**) and protein (**e**, **g**, **i**). **j**, **k** Knockdown of *JUN* partially abrogated SMARCA4-mediated induction of cyclin D1 mRNA (**j**) and protein (**k**) expression in H1703 cells. **l** Proposed model showing that SMARCA4 directly regulates *CCND1* and also upregulates *JUN* which positively regulates *CCND1*. Two-tailed *t*-test. Error bars represent mean ± s.d., ***$p < 0.001$, ****$p < 0.0001$

## Methods

**Cell culture and viral transduction**. Lung cancer cell lines were cultured in RPMI with 7% fetal bovine serum (FBS), 1% penicillin/streptomycin and 2 mM L-glutamine. Cells were maintained at 37 °C and 5% $CO_2$ and regularly tested for Mycoplasma using Mycoalert Detection Kit (Lonza). All cell lines came directly from ATCC or have been validated by short tandem repeat (STR) profiling.

All experiments with ectopic expression and short hairpin RNA (shRNA) knockdown were performed using lentiviral constructs (see Plasmids). Lentiviral transduction was performed using the protocol as described at http://www.broadinstitute.org/rnai/public/resources/protocols. For lentiviral vector related work, cells (30 h post infection) were selected in puromycin or blasticidin for 2–4 days and harvested immediately for the experiments.

**Compounds and antibodies**. Palbociclib (S1116) and Abemaciclib (S7158) were purchased from Selleck Chemicals (Houston, Texas, USA). Antibodies against HSP90 (H-114), cyclin D1 (A12), CDK6 (C-21), CDK4 (DCS-35), p16 (C-20), p21 (H164), cyclin E (HE12), c-Jun (G4) and c-Fos (E8) were from Santa Cruz Biotechnology; antibodies against cyclin D2 (D52F9) and p-RB (S795) were from Cell Signaling; antibody against SMARCA4 were from Bethyl Laboratories (A300-813A). Antibody against Rb (554136) was from BD Pharmingen. Cyclin D3 (ab28283) antibody was from Abcam. Antibody against SMARCA4 was used with 1:5000 dilution and all others with 1:1000 dilution. Antibodies for IHC and ChIP are listed in the corresponding method section below.

**Plasmids**. Individual shRNA vectors used were from the Mission TRC library (Sigma) provided by Genetic Perturbation Service (GPS) of Goodman Cancer Research Center and Biochemistry at McGill University: sh*CCND1*#1 (TRCN0000295876), sh*CCND1*#2 (TRCN0000288598), sh*SMARCA4* (TRCN0000015552), sh*SMARCA2*#1 (TRCN0000358828), sh*SMARCA2*#2 (TRCN0000020329), sh*RB*#1(TRCN0000040163), sh*RB*#2(TRCN0000288710), sh*E2F1*#1(TRCN0000000249), sh*E2F1*#2(TRCN0000000252), sh*E2F1*#3 (TRCN0000039659), sh*JUN*#1(TRCN0000039590), sh*JUN*#2(TRCN0000010366), sh*FOS*(TRCN0000273941). pLX304-*GFP*, pLX304-*CCND1*, pLX317-*GFP* were obtained from TRC3 ORF collections from TransOMIC and Sigma provided by GPS. pReceiver-Lv120, pReceiver-Lv120-*SMARCA4* pReceiver-Lv151, pReceiver-Lv151-*SMARCA2* and pReceiver-Lv105-*SMARCA2* were purchased from GeneCopoeia.

**Colony formation assays**. Since different cell lines have variable proliferation rates and sizes, plating densities for each line were first optimized to allow about 2 weeks of drug treatment, before cells reach 90% confluency in 6-well plates. Single-cell suspensions of all cell lines were then counted and seeded into 6-well plates with the densities predetermined ($2–8 \times 10^4$ cells per well). Cells were treated with vehicle control or drugs on the next day and culture medium was refreshed every 3 days for 10–14 days in total. At the endpoints of colony formation assays, cells were fixed with 3.75% formaldehyde, stained with crystal violet (0.1%w/v) and photographed. All relevant assays were performed independently at least three times.

**Cell viability assays**. Cultured cells were seeded into 96-well plates (200–2000 cells per well). At 24 h after seeding, serial dilutions of CDK4/6 inhibitors were added to cells to final drug concentrations ranging from 0.0026 to 10 μM. Cells were then incubated for 5–7 days and cell viability was measured using the CellTiter-Blue viability assay (Promega). Relative survival in the presence of CDK4/6 inhibitors was normalized to the untreated controls after background subtraction.

**Protein lysate preparation and immunoblots**. Cells were first seeded in normal medium without inhibitors. After 24 h, the medium was replaced with fresh medium containing the inhibitors as indicated in the text. After the drug stimulation, cells were washed with cold phosphate-buffered saline (PBS), lysed with

protein sample buffer and processed with Novex® NuPAGE® Gel Electrophoresis Systems (Invitrogen). HSP90 serves as loading control. Uncropped western blots for the most important experiments are displayed in Supplementary Fig. 15.

**Cell cycle analysis**. H1703 and H1299 cells were treated with palbociclib at indicated concentrations for 24 h before harvesting. Cells were then washed with PBS containing 1% FBS and fixed with 1 ml cold 70% ethanol for 30 min on ice. After washing twice with PBS, cells were treated with 25 μg ml$^{-1}$ Ribonuclease A and stained with 50 μg ml$^{-1}$ propidium iodide solution for 10 min. Guava easy-Cyte™ HT System (Millipore Corporation) was used to analyze cell cycle for the stained cells based on the manufacturer's instructions.

**Annexin V staining**. H1299 and H1703 cells in 96-well plates were treated with palbociclib in medium containing IncuCyte® Annexin V Reagent for Apoptosis (Essen Bioscience, Catalog numbers: 4641). Cells treated with hydrogen peroxide serve as positive controls in these experiments. IncuCyte® live-cell analysis imaging system was used to record 4 images every 2 h. Images were analyzed by IncuCyte® Zoom (2016B) software and annexin V-positive cells were normalized to phase contrast confluency for each well.

**Assay for transposase-accessible chromatin sequencing**. ATAC-seq experiments in H1703 cells were performed exactly as previously described[41]. Briefly, 50,000 cells were harvested after treating with DNase (Worthington cat. no. LS002007) at a final concentration of 200 U per ml. Cell pellets were resuspended in ATAC-Resuspension Buffer (10 mM Tris-HCl pH 7.4, 10 mM NaCl, 3 mM $MgCl_2$) containing 0.1% NP-40, 0.1% Tween-20 and 0.01% digitonin and incubated on ice for 3 min before adding 1 ml of cold ATAC-Resuspension Buffer containing 0.1% Tween-20 but no NP-40 or digitonin. Nuclei were pelleted, resuspended in 50 μl of transposition mixture (25 μl 2× TD buffer (20 mM Tris-HCl pH 7.6, 10 mM $MgCl_2$, 20% dimethyl formamide, Illumina cat. no. 15027866), 2.5 μl TDE1 transposase (100 nM, Illumina cat. no. 15027865), 16.5 μl PBS, 0.5 μl 1% digitonin, 0.5 μl 10% Tween-20, 5 μl $H_2O$) and incubated at 37 °C for 30 min in a thermomixer with 1000 RPM mixing. After cleaning with Zymo DNA Clean and Concentrator-5 Kit (Zymo Research cat. no. D4014), eluted DNA was pre-amplified for 5 cycles using NEBNext 2x MasterMix (NEB cat. no. M0541). Additional amplification cycles were added based on qPCR amplification[42]. PCR reactions were then purified and quantified with Quant-iT™ PicoGreen™ dsDNA Assay Kit (ThermoFisher cat. no. P7589) before sequencing. The primers used for amplification were previously published[43] and are as follows:

Ad1_noMX (forward): AATGATACGGCGACCACCGAGATCTACACTCGTCGGCAGCGTCAGAT GTG;

Ad2.1_TAAGGCGA (reverse, index 1): CAAGCAGAAGACGGCATACGAGATTCGCCTTAGTCTCGTGGGCTCGGAG ATGT;

Ad2.2_CGTACTAG (reverse, index 2): CAAGCAGAAGACGGCATACGAGATCTAGTACGGTCTCGTGGGCTCGGAG ATGT;

Ad2.3_AGGCAGAA (reverse, index 3): CAAGCAGAAGACGGCATACGAGATTTCTGCCTGTCTCGTGGGCTCGGAG ATGT;

Ad2.4_TCCTGAGC (reverse, index 4): CAAGCAGAAGACGGCATACGAGATGCTCAGGAGTCTCGTGGGCTCGGA GATGT.

**Chromatin immunoprecipitation and sequencing**. Cells were fixed in complete media with 1% formaldehyde for 10 min at room temperature or 0.3% formaldehyde for 30 min at 4 °C then quenched by addition of 0.125 M glycine for 5 min at room temperature and 15 min on ice. Fixed cells were then pelleted and washed once with 1× PBS before snap-freezing on dry-ice. Antibodies against E2F1 (Millipore, 05-379) and SMARCA4 (Abcam, ab110641) were used for ChIP

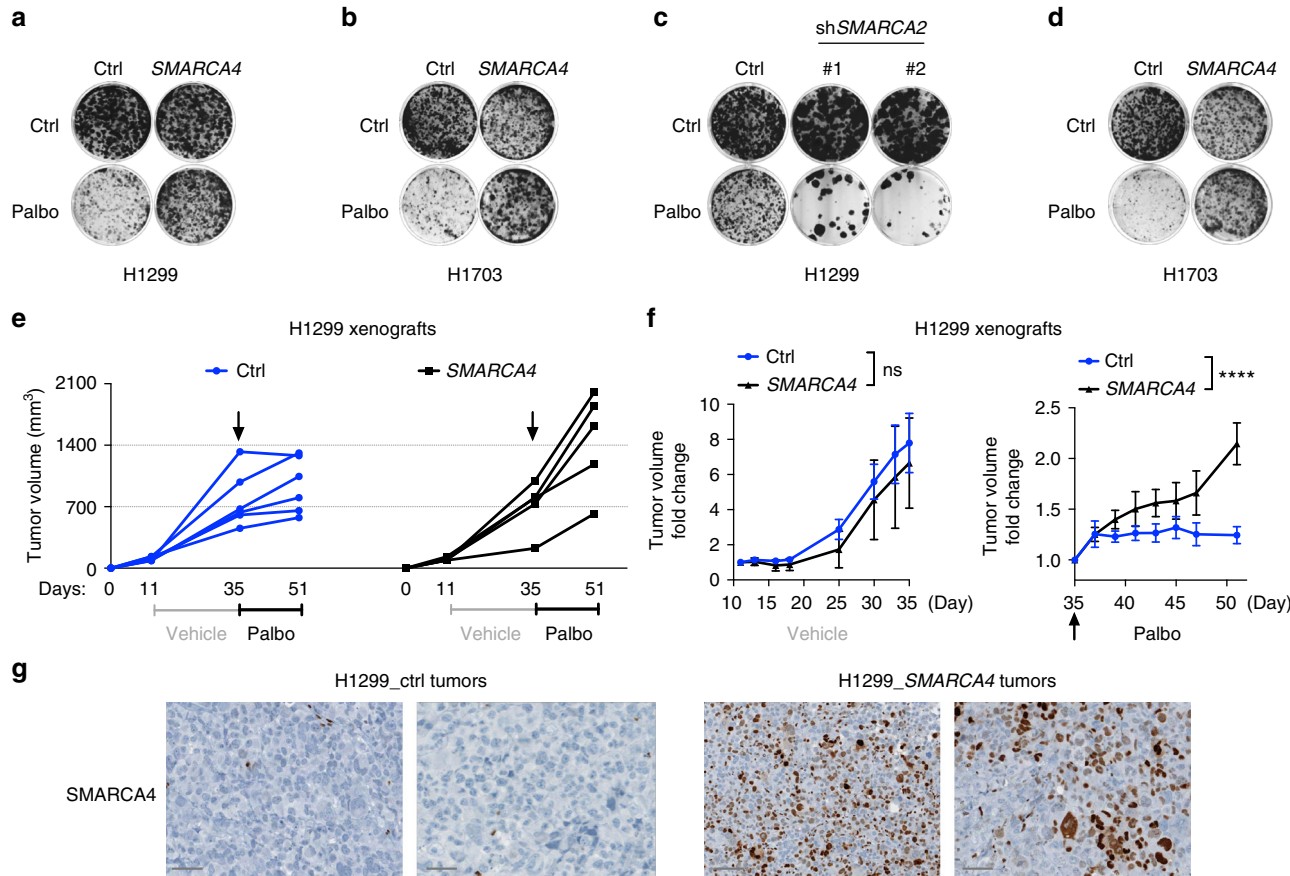

**Fig. 6** SMARCA4 loss is synthetic lethal with cyclin-dependent kinase 4/6 (CDK4/6) inhibition in non-small cell lung cancer (NSCLC). **a, b** SMARCA4 restoration in SMARCA4-deficient cell lines confers drug resistance to palbociclib. Colony formation assays of H1299 (**a**) and H1703 (**b**) cells expressing vector control or *SMARCA4* and treated with palbociclib (H1299, 300 nM; H1703, 100 nM). **c** SMARCA2 knockdown in SMARCA4-deficient H1299 cells sensitizes cells to palbociclib treatment. Colony formation assay of H1299 cells expressing pLKO control or SMARCA2 short hairpin RNAs (shRNAs) and treated with palbociclib. **d** SMARCA2 restoration in SMARCA4/2-dual deficient cells H1703 confers resistance to palbociclib. Colony formation assay of H1703 cells expressing vector control or *SMARCA2* and treated with palbociclib. **e–g** Resistance to palbociclib after restoration of SMARCA4 is also observed in mouse xenograft models using an isogenic cell pair of H1299 cells expressing vector control or *SMARCA4*. **e** Tumor volume evolution during the course of the experiment in H1299 xenograft models expressing vector control (left) or *SMARCA4* (right). **f** Tumor volume fold change during the establishment phase (left) and during palbociclib treatment (right) in the same models. **g** Immunohistochemistry (IHC) analysis of SMARCA4 in the representative endpoint tumors of H1703 control or SMARCA4-restored from above. Bar 50 μm. Error bars represent mean ± standard error of mean (s.e. m.); two-way analysis of variance (ANOVA), ****$p < 0.0001$

experiments by following a protocol using MNase[44]. For histone H3 acetyl K27 and histone H3 tri-methyl K4 ChIP in H1703 cells, SMARCA2 ChIP in H1299 cells and SMARCA4 ChIP in H1915 cells, cell pellets were lysed in three successive buffers for 10 min each at 4 °C while rotating end-over-end (LB1: 50 mM HEPES-KOH pH 7.5, 120 mM NaCl, 1 mM EDTA, 10% glycerol, 0.5% NP-40, 0.25% Triton X-100, LB2: 10 mM Tris-HCl pH 8.0, 200 mM NaCl, 1 mM EDTA, 0.5 mM EGTA, LB3: 10 mM Tris-HCl pH 8.0, 100 mM NaCl, 1 mM EDTA, 0.5 mM EGTA, 0.1% Na-Deoxycholate, 0.5% N-lauroylsarcosine). Lysates were then sonicated with a Branson450D cup-horn system to produce chromatin fragments between 100 and 600 bps. Triton X-100 was added to cell lysate and then centrifuged at $20,000 \times g$ for 15 min at 4 °C to pellet debris. Supernatant equivalent to 10 million cells brought to a final volume of 500 μl using LB3 was used for each immunoprecipitation. A quantity of sonicated chromatin was set aside as 10% input. Then, 5 μg IgG (Abcam ab37415), αSMARCA4 (Bethyl A300-813A), αSMARCA2 (Cell Signaling D9E8B) and 2 μg of histone H3 acetyl K27 (Abcam, ab4729), histone H3 tri-methyl K4 (Abcam, ab8580) antibodies were added to the lysate for overnight incubation at 4 °C. Protein G Magnetic Dynabeads® (ThermoFisher Scientific) were used for pulldown. Immunoprecipitated chromatin bound to dynabeads was washed with four successive buffers (LSB: 20 mM Tris-HCl pH 8.0, 150 mM NaCl, 0.1% SDS, 1% Triton X-100, 2 mM EDTA; MSB: 20 mM Tris-HCl pH 8.0, 250 mM NaCl, 0.1% SDS, 1% Triton X-100, 2 mM EDTA; LiCl wash: 10 mM Tris-HCl pH 8.0, 250 mM LiCl, 0.5% NP-40, 0.5% Na-deoxycholate, 1 mM EDTA; 1× TE: 10 mM Tris-HCl pH 8.0, 1 mM EDTA). Chromatin was then eluted from dynabeads in 150 μl EB (50 mM Tris-HCl pH 8.0, 10 mM EDTA, 1% SDS) by incubating at 65 °C for 30 min. Immunoprecipitated samples and input were incubated

overnight at 65 °C to denature formaldehyde crosslinking. Samples were then treated with RNaseA (ThermoFisher Scientific) followed by proteinase K (Sigma Aldrich) before phenol/chloroform extraction. DNA was precipitated using 5 M NaCl, glycoblue (Ambion) and 100% absolute ethanol overnight at −80 °C. DNA was pelleted by centrifuging at $20,000 \times g$ for 30 min at 4 °C followed by a 70% ethanol wash. Final DNA pellet was resuspended in 50 μl of 1× TE buffer and placed in speedvac for 3 min. All ChIP-sequencing (ChIP-seq) libraries were built with NEBNext Ultra II DNA Library Prep Kit for Illumina (New England Biolabs) by following the manufacturer's instructions.

**ATAC-seq and ChIP-seq data analysis**. ATAC-seq data were mapped to the hg19 genome using bowtie[45], with the parameters -X 2000 -m 1 -p 4. ChIP-seq data were mapped to hg19 using bowtie2 with default parameters. Duplicate reads were removed using samtools[46]. ATAC-seq peaks were calculated using MACS2[47], and overlap between peaks was determined using bedtools[48]. Location of peak's nearest gene was determined using the annotatepeaks function of Homer[49]. Metaplots and heat maps of ChIP and ATAC-seq data over peak sets were generated using ngsplot[50]. For visual display, SMARCA4 ChIP-seq data were smoothed using the bamCoverage function of deepTools, with a smoothLength of 300[51]. ATAC-seq and ChIP-seq data were imaged using IGV[52].

**RNA isolation and qRT-PCR**. Cells were first seeded and then harvested for RNA isolation using Trizol (Invitrogen) the next day. Synthesis of complementary DNAs (cDNAs) using Maxima First Strand cDNA Synthesis Kit (Thermo Scientific) and

real-time quantitative reverse transcription PCR (qRT-PCR) assays using SYBR® Green master mix (Roche) were carried out according to manufacturer protocols. Relative mRNA levels of each gene shown were normalized to the expression of the housekeeping gene *GAPDH*. The sequences of the primers for qRT-PCR assays are as follows:

*GAPDH*_Forward (Fwd), AAGGTGAAGGTCGGAGTCAA;
*GAPDH*_Reverse (Rev), AATGAAGGGGTCATTGATGG;
*CCND1*_Fwd, GGCGGGATTGGAAATGAACTT;
*CCND1*_Rev, TCCTCTCCAAAATGCCAGAG;
*JUN*_Fwd, TTCTATGACGATGCCCTCAACGC;
*JUN*_Rev, GCTCTGTTTCAGGATCTTGGGGTTAC.

The primers used for the chromatin immunoprecipitation were designed based on publicly available Encode ChIP-seq tracks of SMARCA4 and are as follows:

*CCND1* Promoter Fwd, CCGGAATGAAACTTGCACAGG;
*CCND1* Promoter Rev, AGACGGCCAAAGAATCTCAGC;
*CCND1-13kb* Fwd, AAGTCACTCTTCCGTAGAGC;
*CCND1-13kb* Rev, GGCACCTGGACCTTCAACAC;
*CCND3* Promoter Fwd, CCTCCCATTTTGCTTCTCGG;
*CCND3* Promoter Rev, TGAGTCATTACATCGTGAGG;
*CCND1-31kb* Fwd, TTGCTGCTCTGCCACTCTTAC;
*CCND1-31kb* Rev, CCATCTGTCAGTTCATGTCAAGC.

**Mouse xenografts and in vivo drug studies**. For in vivo drug studies, palbociclib (SelleckChem, S1116) was resuspended in 50 mM sodium L-lactate (Sigma Aldrich) buffer (pH = 4.0) at a concentration of 15.75 mg ml$^{-1}$ (150 mg kg$^{-1}$ dose for a 21 g mouse in a volume of 200 μl) and stored at −80 °C. Tubes were thawed overnight at 4 °C.

Animal experiments were carried out according to standards outlined in the *Canadian Council on Animal Care Standards (CCAC)* and the *Animals for Research Act*, R.S.O. 1990, Chapter c. A.22, and by following internationally recognized guidelines on animal welfare. All animal procedures (Animal Use Protocol) were approved by the Institutional Animal Care Committee according to guidelines of the Canadian Council of Animal Care. All animal experiments were carried out at the Research Institute of McGill University Health Center, using 8–12-week-old female YFP/SCID mice (bred in house). For the tumor model, single-cell suspension was created by dissociating a sufficient number of sub-confluent flasks of cells to produce 1 million cells for H1299 and 3 million cells for H1703 in 200 μl of Matrigel HC in a 50:50 ratio (Corning Matrigel HC, cat. no. 354428, VWR, Mississauga, Canada). The tumor cell suspension was subcutaneously injected into the left flank of each SCID mouse. When tumor volumes ($V = (H × W^2)/2$) reached ~60 mm$^3$ (11 days post inoculation), which was assigned as day 1, the mice were entered into the treatment regimen (200 μl p.o. × 24 days). Mice were randomly allocated to vehicle control (50 mM sodium L-lactate buffer, pH 4.0) or the treatment group (150 mg kg$^{-1}$ palbociclib). Mice were housed in groups of 3–5, with each group consisting of both vehicle control and treatment animals matched for tumor size on day 1 of treatment. All gavage treatments were carried out using sterile straight 22-gauge, 38.1 mm stainless steel feeding tubes (Harvard Apparatus, QC). Tumor progression was monitored and measurements using digital calipers (VWR International) were recorded twice weekly. The persons who performed all the tumor measurements and the IHC analysis for the endpoint tumor samples were blinded to the treatment information.

**Patient tumor samples**. Gene expression microarray data for 83 lung adenocarcinoma tumors collected under informed consent at the British Columbia Cancer Agency (Vancouver, BC) were generated using Illumina HumanWG-6 v3.0 arrays. Bead-level data were processed with the MBCB algorithm[53], quantile-normalized and log2-transformed[54]. These data and additional information are available through the Gene Expression Omnibus (Accession Number = GSE75037). TCGA data (RNA-seq RSEM values) for 230 lung adenocarcinomas were downloaded from the MSKCC cBioPortal. Associations between *CCND1* and *SMARCA4* expression were assessed using Pearson correlations with $p < 0.01$ considered significant.

For the immunohistochemistry studies, patient samples ($n = 110$) of the NSCLC were provided by the tissue bank of the National Center for Tumor Diseases (NCT, Heidelberg, Germany) in accordance with the regulations of the tissue bank and the approval of the ethics committee of Heidelberg University (NCT Heidelberg, Ref.no. 206/2005; 207/2005) was obtained. Studies on resected lung adenocarcinoma patient tumors ($n = 100$) were approved by the ethics boards at the at McGill University Health Center (F11HRR–17212).

**Immunohistochemistry**. For mouse xenografts, 4 μm thick sections from formalin-fixed, paraffin-embedded tissue were cut, deparaffinized and stained using an IntelliPath automated immunostainer (Biocare Medical). The protocol included an antigen retrieval treatment in Diva Decloaker RTU (Biocare Medical) for 10 min followed by incubation with the primary antibody (phosphoRB, Cell Signaling, 9308, 1:200 dilution; KI67, Abcam, 16667, 1:100 dilution; SMARCA4, Abcam, 110641, 1:100 dilution) for 1 h at room temperature. Incubation was followed by detection using a Goat anti Rabbit horseradish peroxidase (Dako) and 3,3′-diaminobenzidine (DAB; Dako). The slides were digitalized using an Aperio scanner.

The mitotic index was measured by counting the mitotic active cells in 10 high power fields (400×) of the hematoxylin and eosin (H&E)-stained tumor slides.

For patient tumor samples, national Center for Tumor Diseases (NCT) cohort: IHC was performed on 3 μm sections cut from paraffin blocks using a fully automated system ("Benchmark XT System", Ventana Medical Systems Inc, 1910 Innovation Park Drive, Tucson, Arizona, USA). IHC was performed on the TMAs using anti-SMARCA4 (anti-BRG1 antibody (clone EPNCIR111A, dilution, 1:100 dilution, Abcam, Cambridge, UK), anti-p16 (clone G175-405, 1:20 dilution, BD Pharmingen™) and anti-RB1 (G3-245, 1:100 dilution, BD Pharmingen™) antibodies. Positive and negative controls were used throughout. For assessment of SMARCA4, SMARCA2 and RB1, unequivocally absent staining in the nuclei of viable tumor cells as opposed to strong staining in background stromal cells was considered IHC negative. Expression in the tumor cells that is equivalent to the staining of non-neoplastic cells in the background was considered IHC positive. For p16, only nuclear or combined nuclear and cytoplasmic staining was considered specific. For cyclin D1, nuclear staining results were analyzed using H-score using Zeiss microscope at a ×100 magnification. Positive cells were analyzed according to the staining intensity on a scale of 0–3 (0 = negative, 1 = weak, 2 = moderate, 3 = strong). H-scores were calculated as the sum of the percent of cells at each intensity (Pi) multiplied by the intensity score (i). H-score = Σ (Pi(i)) × 100. Score values range between 0 and 300.

**McGill University Health Center (MUHC) cohort**. Tissue microarrays (TMAs) were constructed from these lung adenocarcinoma samples and are comprised of 4 mm cores from the selected paraffin-embedded tissue blocks. The 4 μm thick sections from these TMAs were cut, deparaffinized and stained using the Bench-Mark Ultra system (Ventana Medical Systems Inc). Heat-induced epitope retrieval (HIER) was performed with Ultra Cell Conditioning Solution (CC1) for 32 min at 100 °C, followed by blocking with antibody diluent with casein (Ventana), and 32 min of incubation at 36 °C with the mouse monoclonal antibody against SMARCA4 (clone sc-17796, dilution 1:100, Santa Cruz Biotechnology). For cyclin D1, HIER was performed in CC1 for 48 min at 100 °C followed by a 48 min of incubation at 36 °C with the rabbit monoclonal antibody against cyclin D1 (clone SP4, dilution 1:100, Cell Marque–Sigma Aldrich, Rocklin, California). After primary antibody incubation, detection was performed using the default OptiView DAB protocol as per the manufacturer's directions (Ventana). The slides were digitalized using an Aperio scanner. The IHC results were interpreted and scored as described above. Cores with low tumor cellularity and artifacts were not included in the analysis.

**Statistical analysis**. Statistical significance was calculated by two-tailed Student's $t$-test, two-way analysis of variance (ANOVA) or non-parametric Mann–Whitney test accordingly. Prism 6 software was used to generate graphs and statistical analyses. Error bars represent mean ± standard deviation (s.d.) or standard error of mean (s.e.m.); *$p < 0.05$, **$p < 0.01$, ***$p < 0.001$, ****$p < 0.0001$.

**Reporting summary**. Further information on experimental design is available in the Nature Research Reporting Summary linked to this article.

## Data availability

All high-throughput sequencing data sets have been deposited in the Gene Expression Omnibus (GEO) under accession no. GSE121755. SMARCA4 ChIP-seq in H1299 cells was obtained from published sources[34]. FOS and JUN ChIP-seq data of HUVEC cells (GSM935585, GSM935278) was taken from the ENCODE database[55]. All other data in this study are available through contacting the corresponding authors on reasonable request.

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

## Acknowledgements

This work was supported by the Canadian Institutes of Health Research (CIHR) grants, MOP-130540 (S.H.), PJT-156233 (S.H.), FDN-148390 (W.D.F.), FDN-143322 (J.R.) and MOP-142451 (J.D.), a Natural Sciences and Engineering Research Council of Canada (NSERC) grant RGPIN-2018-04856 (W.P.), and a Cancer Research Society (CRS) grant 2957 (J.S.). Y.X. is supported by Rolande & Marcel Gosselin Graduate Studentship and Charlotte & Leo Karassik Foundation Oncology PhD Fellowship, J.S. is supported by MUHC Research Institute Foundation (F. Ann Birks Fellowship #8195 and Ray Chiu Research Award #2671) and American Association of Thoracic Surgeons, W.L. is supported by a Terry Fox Research Institute (TFRI) New Investigator Award, J.R. is supported by a Jack Cole Chair and S.H. is supported by a Canadian Research Chair (CRC) in Functional Genomics.

## Author contributions

Y.X., B.M., Z.F., X.Q.D.W., C.L., T.K., X.Z., G.M. and L.S. performed experiments. Y.X., W.W.L. and W.A.P. performed statistical analyses. E.H., S. J., S.C.-B., A.V.G., W.W.L., W.A.P. and J.D.S. contributed samples and technical support and provided advice. P.F., R.R., S.V., D.M., A.R.J., M.-C.G and A.A. provided pathology expertise. A.A., W.A.P., J. D., J.R., W.D.F and S.H. supervised the experiments. Y.X., W.D.F. and S.H. wrote the manuscript. S.H. oversaw the study. All authors read and approved the final manuscript.

## Additional information

**Competing interests:** The authors declare no competing interests.

