## [Peer Review File · Nature Communications]

Reviewers' comments:

Reviewer #1 (Remarks to the Author):

In this manuscript Xue et al explore the role of SMARCA4 loss, alone or in combination with SMARCA2 loss, promoting synthetic lethality with CDK4/6 inhibition. First, they demonstrate that SMARCA4 or combined SMARCA4/2 deficient cell lines exhibit reduced Cyclin D1 expression both at the protein and mRNA levels. Next, they demonstrate in colony forming assays that palbociclib IC50 is under 100 nM for these lines, which is similar to the sensitivity of KRAS mutant cells. They further demonstrate that Cyclin D1 over-expression rescues palbo sensitivity in SMARCA4 deficient cells, while Cyclin D1 suppression enhances sensitivity in SMARCA4/2 intact cells. They also show that SMARCA4 reconstitution rescues Cyclin D1 and impairs sensitivity to palbo, and that combined modulation SMARCA2 further regulates Cyclin D1 expression and palbo sensitivity. Finally, they demonstrate in vivo relevance, correlating SMARCA4 and especially dual loss of SMARCA2 with Cyclin D downregulation in human tumors, and robust sensitivity of SMARCA4 and SMARCA4/2 cell line xenografts to palbo.

In general this is a well conducted study. My primary critique is that the mechanistic connection between SMARCA4 and cyclin D regulation remains unclear, and could be confounded by cell cycle phenotypes. Only chromatin IP is shown in the supplementary information, without robust controls, and no cell cycle effects are considered.

Major Critiques

1. At a minimum whether SMARCA4 directly regulates Cyclin D1 expression should be more firmly established. It would be nice to see additional specificity controls for the ChIP, and this data should be in the main figure – for example since the SMARCA4 Ab results could still be non-specific it would be nice to see that SMARCA4 ChIP in H1299 (which lacks SMARCA4) fails to enrich for the Cyclin D locus.
2. Since E2F1 directly regulates Cyclin D1 expression do SMARCA4/2 promote E2F1 accessibility? If enhanced E2F1 binding to the Cyclin D1 locus were observed in SMARCA4/2 intact cells versus null cells this would be much stronger evidence that the findings are direct, and should again be included in the main figure since they provide a clear mechanistic connection.
3. Do low Cyclin D1 levels simply correlate with reduced proliferation rate in SMARCA4/2 lines? I suspect this not to be the case, given that KRAS mutant lines such as A549 and H23 proliferate rapidly, but it would be helpful to compare doubling times for at least a subset of lines, or more importantly clarify the plating density and time to confluence for each control treated cell line in the colony formation assays.
4. Is the synthetic lethality observed related to true lethality (ie apoptosis) or enhanced G1 arrest? This should be explored in at least a subset of cell lines treated with palbo.

Reviewer #2 (Remarks to the Author):

Summary: In this study, Xue and colleagues assess the impact of SMARCA4 and SMARCA2 deficiency on the sensitivity of human non-small cell lung cancer cell lines to CDK4/6 inhibition. The authors demonstrate a reasonably-well-correlated increased sensitivity to the CDK4/6 inhibition in cell lines that lack expression of SMARCA4 and a further increase when cells lack both SMARCA4 and SMARCA2 expression. Mechanistically the authors suggest a model where absence of SMARCA4/2 leads to the diminution of CCND1 mRNA and protein expression to explain the differential sensitivity to CDK4/6 inhibition. The experiments were performed in numerous cell lines (~20 independent lines). In a subset of these lines, the authors also include the requisite knockdown and “add back” experiments to establish a likelihood that these are causal relationships. Chromatin IP data suggest that SMARCA4/2 can bind to the CCND1 promoter, and

that this binding is correlated with positive gene expression from the locus. The authors also provide relatively weak positive correlations of SMARCA4/2 expression with that of CCND1 in human cancer databases. Finally the authors repeat the correlation of SMARCA4/2 deficiency and palbociclib (CDK4/6 inhibition) sensitivity in xenograft assays showing that one two cell lines (one that has low SMARCA4 and one the at has low SMARCA4/2 expression) are sensitive to palbociclib.

Minor points:

- * It is surprising that loss of SMARCA4/2 expression is so widespread in these cell lines given the low frequency of mutation for SMARCA4 (and 2) in TCGA and other databases. What is the explanation for this?

Major Points:

- * In the xenograft experiments: No cell lines with higher levels of SMARCA4/2 are shown to demonstrate differential sensitivity to CDK4/6 inhibition in vivo. No experimental perturbations of SMARCA4/2 levels are performed to demonstrate that sensitive lines can be made resistant or that resistant lines can be made sensitive. This is necessary.

- * The authors show an array of protein blotting for relevant components of the RB pathway.

- * Why are the levels of Rb phosphorylation similar across the ~20 cell lines despite the differential expression of Cyclin D1? If this is the true driver of the phenotype I would presume that Rb phosphorylation would be different unless an alternative compensatory mechanism is in play.

- * Along the same lines, CDK2 is a major player in the phosphorylation of RB. It would be insightful to know whether CDK2 expression (or better yet activity) was altered amongst these cell lines to explain the maintenance of the observed Rb phosphorylation.

- * This work was submitted as a companion paper to another focused on small cell ovarian cancers that have SMARCA4 loss. As such, this manuscript provides little to no additional mechanistic insights beyond that manuscript. For example, because SMARCA4 is not a transcription factor but instead a member on the BAF and PBAF chromatin remodeling complexes it would be important to determine whether the chromatin structure is altered at the CCND1 locus, or other loci, in cells that have lost expression of SMARCA4/2? How does the add back or acute loss of SMARCA4/2 affect that chromatin structure?

- * Despite the potential for clinical impact, I would expect that insights into the chromatin structure an function would be needed to support publication of an additional manuscript linking the SMARCA4 genotype with palbociclib sensitivity phenotype. Additional data in another cancer type beyond a very rare ovarian cancer subtype does not warrant companion publication. As is, these data are better suited for a specialized translational research oriented cancer journal.

REVIEWERS' COMMENTS:

Reviewer #1 (Remarks to the Author):

In general this is a well conducted study. My primary critique is that the mechanistic connection between SMARCA4 and cyclin D regulation remains unclear, and could be confounded by cell cycle phenotypes. Only chromatin IP is shown in the supplementary information, without robust controls, and no cell cycle effects are considered.

1. At a minimum whether SMARCA4 directly regulates Cyclin D1 expression should be more firmly established. It would be nice to see additional specificity controls for the ChIP, and this data should be in the main figure – for example since the SMARCA4 Ab results could still be non-specific it would be nice to see that SMARCA4 ChIP in H1299 (which lacks SMARCA4) fails to enrich for the Cyclin D locus.

During the course of this revision, a SMARCA4 ChIP-seq study in H1299 cells engineered to express inducible exogenous SMARCA4 became available (Lissanu Deribe et al, 2018; PMID: 29892061). This study used an anti-SMARCA4 antibody (Abcam), which is different than the one that we used previously (Bethyl). We analyzed this data set and observed SMARCA4 occupancy at the *CCND1* promoter only upon SMARCA4 induction in H1299 cells (included in **new Fig. 5a**).

In addition, we also conducted additional ChIP-seq experiments in H1703 cells before and after SMARCA4 restoration using the new antibody and obtained the similar results (**new Fig. 5a**).

Furthermore, as shown in the accompanying SCCOHT manuscript (*page 63, Supplemental Figure 9*), publicly available data sets from 8 human cell lines of different tissue origins also show consistent SMARCA4 occupancy at the *CCND1* promoter.

Thus, these independent data support that SMARCA4 directly regulates cyclin D1.

2. Since E2F1 directly regulates Cyclin D1 expression do SMARCA4/2 promote E2F1 accessibility? If enhanced E2F1 binding to the Cyclin D1 locus were observed in SMARCA4/2 intact cells versus null cells this would be much stronger evidence that the findings are direct, and should again be included in the main figure since they provide a clear mechanistic connection.

We have performed the suggested E2F1 ChIP experiment and did observe enhanced E2F1 binding to the *CCND1* promoter region when SMARCA4 was restored in H1703 cells (**new Supplementary Fig. 11**), suggesting increased chromatin accessibility. On the other hand, E2F1 binding may not explain the *CCND1* induction by SMARCA4 as Watanabe G, *et al* (1998; PMID: 9584162) have shown that E2F1 binding suppresses *CCND1*. Indeed, we observed that E2F1 knockdown results in slight upregulation of cyclin D1 in H1703 cells (**new Supplementary Fig. 11**).

Nevertheless, it is an important point that SMARCA4-regulated chromatin accessibility is a key mechanism by which cyclin D1 is regulated. We thank the referee for raising this important comment. We have conducted additional experiments further establishing this regulation and

providing substantial new mechanistic insights for the regulation of *CCND1*. Please see our response to *the major point #3 of the second reviewer* below for details.

3. Do low Cyclin D1 levels simply correlate with reduced proliferation rate in SMARCA4/2 lines? I suspect this not to be the case, given that KRAS mutant lines such as A549 and H23 proliferate rapidly, but it would be helpful to compare doubling times for at least a subset of lines, or more importantly clarify the plating density and time to confluence for each control treated cell line in the colony formation assays.

As suggested, we have determined the doubling time of all cell lines using IncuCyte live cell imaging system (**new Supplemental Fig. 3**). As shown in the **Reviewer Only Figure 1** below, the proliferation rate does not correlate with the cyclin D1 expression or drug sensitivity. In addition, we have clarified the plating density and time in the **Methods**.

Review only figure 1

4. Is the synthetic lethality observed related to true lethality (ie apoptosis) or enhanced G1 arrest? This should be explored in at least a subset of cell lines treated with palbo.

We have included additional data to clarify this. Palbociclib treatment in SMARCA4-deficient H1299 and H1703 cells induces strong G1 cell cycle arrest (**new Figure 1e, f**), but not cell death as indicated by the lack of Annexin V staining (**new Supplementary Fig. 5**).

Reviewer #2 (Remarks to the Author):

Minor points:

** It is surprising that loss of SMARCA4/2 expression is so widespread in these cell lines given the low frequency of mutation for SMARCA4 (and 2) in TCGA and other databases. What is the explanation for this?*

Indeed, it has been reported that 24% of lung cancer cell lines (n=59) carry SMARCA4 mutations (Medlina et al, Hum Mutat, 2008; PMID: 18386774), which is higher than from patient data as pointed out by the reviewer. The authors of this study suggested that “Rather than being an artifact of cells in culture, the lower rates of BRG1 mutations found in primary tumors are probably due to the masking effect of normal cell contamination”. In addition, loss of SMARCA4/2 expression can also result from epigenetic silencing (PMID: 17546055; PMID: 29391527).

Major Points:

** In the xenograft experiments: No cell lines with higher levels of SMARCA4/2 are shown to demonstrate differential sensitivity to CDK4/6 inhibition in vivo. No experimental perturbations of SMARCA4/2 levels are performed to demonstrate that sensitive lines can be made resistant or that resistant lines can be made sensitive. This is necessary.*

We agree with the reviewer that it is important to further establish the role of SMARCA4 in regulating responses to CDK4/6 inhibitors *in vivo*.

While naturally occurred SMARCA4-proficient NSCLC cell lines service as a control for the comparison of drug sensitivity *in vitro*, additional genetic alterations in these cells could lead to different tumor establishment rate, microenvironment etc, which would confound the interpretation of the differential drug response *in vivo*. Therefore, to better address the reviewer’s comment, we have conducted xenografts experiments using two isogenic cell pairs of H1703 and H1299 engineered to only differ in SMARCA4 status.

Our *in vitro* studies show that forced SMARCA4/2 expression in H1703 causes noticeable growth inhibition (**Figure 6b, d**), which may suppress tumor establishment. Therefore, we employed an inducible SMARCA4 system to help establish H1703 xenograft tumors. However, subsequent induction of SMARCA4 expression significantly suppressed the tumor growth and resulted in a strong selection for low/negative SMARCA4 populations (please see **Reviewer Only Figure 2** below). Thus, this model is not suitable for the long-term *in vivo* drug study and was not continued further.

We therefore focused on H1299 model, as these cells can better tolerate forced SMARCA4 expression *in vitro* (**Figure 6a**) and *in vivo* (**new Figure 5e, f, g**). Consistent with our previously described *in vitro* synthetic lethality (**Figure 6a, b**), palbociclib treatment strongly suppressed the growth of SMARCA4-deficient control tumors, but not of SMARCA4-restored tumors (**new Figure 5e, f, g**). These results support that SMARCA4 expression levels modulate drug responses to CDK4/6 inhibition *in vivo*.

Reviewer Only Figure 2

Note that the trend of anti-correlation between tumor size (b) and SMARCA4 IHC signal (c) in the 3 representative endpoint tumors treated with doxycycline to induce SMARCA4 expression.

- * *The authors show an array of protein blotting for relevant components of the RB pathway.*
- * *Why are the levels of Rb phosphorylation similar across the ~20 cell lines despite the differential expression of Cyclin D1? If this is the true driver of the phenotype I would presume that Rb phosphorylation would be different unless an alternative compensatory mechanism is in play.*
- * *Along the same lines, CDK2 is a major player in the phosphorylation of RB. It would be insightful to know whether CDK2 expression (or better yet activity) was altered amongst these cell lines to explain the maintenance of the observed Rb phosphorylation.*

In our accompanying SCCOHT manuscript, we show that reduced cyclin D1 expression limits CDK4/6 kinase activity, which could result in less buffering effect and therefore vulnerability to CDK4/6 inhibition. However, it does not necessarily lead to reduced total RB phosphorylation.

In addition, cell lysates analyzed (such as in **Fig. 1a**) were from a mixture of cell populations at different cell cycle stages, where other kinases are known to phosphorylate RB. Therefore, the levels of RB phosphorylation might not reflect CDK4/6 activity activated by cyclin D1. As suggested by the reviewer, we profiled the expression of CDK2 in our cell line panel. Indeed, we

observed that SMARCA4-deficient cells tend to express higher levels of CDK2 compared to controls cells (**new Supplemental Figure 1**), which might explain the similar RB phosphorylation observed across the cell line panel. The detailed mechanism requires further investigations which are beyond the scope of this study.

** This work was submitted as a companion paper to another focused on small cell ovarian cancers that have SMARCA4 loss. As such, this manuscript provides little to no additional mechanistic insights beyond that manuscript. For example, because SMARCA4 is not a transcription factor but instead a member on the BAF and PBAF chromatin remodeling complexes it would be important to determine whether the chromatin structure is altered at the CCND1 locus, or other loci, in cells that have lost expression of SMARCA4/2? How does the add back or acute loss of SMARCA4/2 affect that chromatin structure?*

We thank the reviewer for the helpful suggestion. This important point regarding SMARCA4-regulated chromatin accessibility was also raised by the first referee. To this end, we have conducted a series of chromatin analyses and functional experiments, providing further insights for the regulation mechanisms (**new Fig. 4, 5** and **new Supplemental Fig. 9, 11-14**). We summarize the key new findings below:

1) ATAC-seq experiments show that re-expression of SMARCA4 or SMARCA2 in H1703 cells both leads to similar global effects on opening of the chromatin landscape (**new Fig. 4a-d**), including the increased chromatin accessibility of the *CCND1* locus (**new Fig. 5a**).

2) H3K27Ac ChIP-seq studies in the same cells show that gain of H3K27Ac in the vicinity of SMARCA4/2-dependent open chromatin (**new Fig. 4e, f**), which include the *CCND1* promoter region (**new Fig. 5a**). Thus, these data further support the enhanced chromatin accessibility of the *CCND1* locus when SMARCA4/2 are restored.

3) Our ATAC-seq results also show that SMARCA4/2 restoration leads to an opening of a putative enhancer region upstream of the *CCND1* locus, containing two strong ATAC-seq peaks (**new Fig. 5a**). Consistent with this, ChIP-Seq signal of H3K27Ac but not H3K4m3 was also strongly increased at these two peaks upon SMARCA4/2 restoration (**new Fig. 5a**). The summit of each peak contains a canonical AP-1 site (**new Fig. 5c**), the motif bound by Fos/ Jun dimers. Analysis of existing ChIP-seq data in human endothelial cell line HUVEC show strong c-Jun/c-Fos binding at this site (GSM935278).

4) Given that c-Jun/c-Fos are known to activate *CCND1* (PMID: 7559524; 17637753), we examined their role in mediating SMARCA4/2 regulation of cyclin D1. We found that c-Jun, but not c-Fos, is upregulated by SMARCA4/2 re-expression in H1703 and H1299 cells (**new Fig. 5 f-k** and **new Supplemental Fig. 14**). Importantly, knockdown of c-Jun or c-Fos partially abrogated SMARCA4-mediated induction of cyclin D1 mRNA and protein expression (**new Fig. 5i, m** and **new Supplemental Fig. 14**), suggesting that SMARCA4 activates cyclin D1 expression in part through c-Jun/c-Fos.

Whether this c-Jun/c-Fos mediated cyclin D1 regulation is through the putative enhancer or/and other mechanism requires further investigation which is beyond the scope of this study.

Together, these new data provide new mechanistic insights demonstrating that SMARCA4/2 SMARCA2/4 loss reduces cyclin D1 expression by a combination of restricting *CCND1* chromatin accessibility and suppressing c-Jun.

** Despite the potential for clinical impact, I would expect that insights into the chromatin structure and function would be needed to support publication of an additional manuscript linking the SMARCA4 genotype with palbociclib sensitivity phenotype. Additional data in another cancer type beyond a very rare ovarian cancer subtype does not warrant companion publication. As is, these data are better suited for a specialized translational research oriented cancer journal.*

We appreciate that the reviewer recognizes the potential impact of our study.

As discussed above, we have included multiple lines of evidence providing new mechanistic insights for the *CCND1* regulation in NSCLC; we have also provided additional *in vivo* data further establishing the connection between SMARCA4 loss and sensitivity to CDK4/6 inhibitors.

In summary, we have substantially improved our manuscript in this revision guided by the helpful suggestions of both reviewers. We hope that you will find this revised manuscript suitable for publication in Nature Communications.

Reviewers' comments:

Reviewer #1 (Remarks to the Author):

The authors have satisfactorily addressed my concerns.

Reviewer #2 (Remarks to the Author):

In this revised manuscript by Xue et al. the authors have addressed the substantive concerns that I had previously raised, although some more thoroughly than others. However, whether the manuscript has made a sufficiently impactful discovery that warrants publication in Nature Communications is the concern. The major points are outlined below with my general comments at the end regarding the suitability for publication in Nature Communications.

Point #1) The authors provide a significant amount of new analyses to look at the mechanisms that SMARCA4/2 employ to regulate CCND1 expression. Previously, a weak ChIP association of SMARCA4 with the CCND1 promoter was presented as the mechanism supporting the model. Now ATAC-seq data has been added to show that forced SMARCA4/2 expression can increase chromatin accessibility at the CCND1 promoter and a putative enhancer sequence upstream of the gene has been identified. Additional H3K27Ac data support the ATAC-seq data and the notion that an enhancer sequence is newly opened and active after SMARCA4 re-expression. Interestingly, the putative new enhancer region contains two AP-1 sequences that large scale orthogonal data sets have previously shown to bind c-Jun. The authors show additional data that SMARCA4 may drive JUN expression by associating with the JUN promoter, increasing chromatin accessibility at that promoter, and inducing higher c-Jun protein levels in the cells. The model supported by the data is that SMARCA4/2 expression opens chromatin structure at the CCND1 promoter and an upstream enhancer, while also inducing expression of c-Jun which binds the newly opened enhancer to ultimately support higher CCND1 expression. CCND1 expression levels in turn dictate sensitivity to CDK4/6 inhibitors. These added data do add mechanistic support overall and an interesting model; though the involvement of typical players regulating cell proliferation (i.e. AP-1 transcription factors) diminishes interest.

Point #2) Data to address the the lack of add back of SMARCA4/2 in NCLC cell lines that naturally lack SMARCA4/2 expression during xenografts has been added. The results are supportive of the overall conclusions, although they were only conducted on one cell line. H1299 cells were made into an isogenic pair (Parental line that is SMARCA4 deficient and one that expresses SMARCA4 from a cDNA). Xenografts derived from this cell line are made resistant to palbociclib by SMARCA4 add back supporting the model.

The authors also show that SMARCA4 add back to H1703 cells can confer palbociclib resistance in vitro. The data in figure 6b, d show a very small diminution of H1703 cell growth after SMARCA4 or SMARCA2 add back and the authors claim in their rebuttal that they were too worried that this would impact the ability of the line to form xenografted tumors. However, the relative difference between the parental and SMARCA4 expressing H1299 (Fig 6a) and H1703 (Fig6b) cells seems pretty similar. I just don't see the logic for ruling out one of the lines based on this and for developing an entirely new system for the H1703 cells where they can induce SMARCA4 expression with dox, especially when this strategy failed to help in the end anyway (Reviewer Figure 2).

The authors have ignored my suggestion that they knockdown SMARCA4/2 in cell lines that have high expression and asses the impact on CDK4/6 sensitivity during xenograft analysis.

Overall this seems like a halfhearted and haphazard approach to address my comment though the data selected for inclusion in the manuscript does address the concern.

General Comments

In the end, the question remains as to what does this manuscript provide that is additional to the companion manuscript? No doubt, it is clinically important to show that SMARCA4/2 deficient NSCLCs have heightened sensitivity to palbociclib. However, this is a “me too” result if published alongside another paper outlining the initial discovery found using an interesting and somewhat unbiased screening approach. The added chromatin remodeling studies are interesting and likely explain the differential expression of CCND1 and sensitivity to CDK4/6 inhibition. However, some insights are limited and only just beginning to be explored by the authors. For example, new Figure 4 offers only a global overview of the impacts on chromatin accessibility and H3K36Ac after SMARCA4/2 overexpression which would be expected to occur after over-expressing a critical component of chromatin regulating complex that was previously absent. Any detailed insight is not revealed until figure 5. Figures 1,2,3 and 6 are derivatives of the companion paper and establish genotype dependent CDK4/6 sensitivity. So the real novelty here comes solely from figure 5 and the insight is that a canonical transcription factor (AP-1) controls proliferation by increasing expression of a canonical cyclin (CCND1). Weighing the insights gained against the relatively weak approaches that rely largely on human cancer cell lines and overexpression studies, my opinion is that this manuscript falls short of the mark.

Reviewer #3 (Remarks to the Author):

I have been asked to review the novel set of data (CHIP-seq and ATAC-seq) that have been added during revision to the manuscript. Overall, the data seem of good quality, but my interpretation is limited by several factors. First of all, the authors do not provide any analytical parameters such as number of decuplicated read per sample, alignment rates etc, to judge if the analysis was done appropriately, I could only see heatmaps and IGV snapshot.

Secondly, there is something rather odd in figure 4f as there is a stark change in the background in the heatmap, almost as if two heatmaps were generated independently and then fused. For figure 5 IGV plots, at the bare minimum there should be a RPKM number for each plot, to make sure the comparison is fair. Finally, in figure 4 it would good to have the heatmaps including the common 40k or so sites to have confidence that the acquired ones are similar in terms of enrichment (it would appear so from the CCND1 and JUN IGV snapshots).

It is also very interesting that almost all the gained ATAC and CHIP signal occur at distal regulatory elements, where the common are much more biased toward promoters. I was wondering if the authors could comment on that.

REVIEWERS' COMMENTS:

Reviewers' comments:

Reviewer #1 (Remarks to the Author):

The authors have satisfactorily addressed my concerns.

We are pleased to see that this reviewer has responded positively.

Reviewer #2 (Remarks to the Author):

In this revised manuscript by Xue et al. the authors have addressed the substantive concerns that I had previously raised, although some more thoroughly than others. However, whether the manuscript has made a sufficiently impactful discovery that warrants publication in Nature Communications is the concern. The major points are outlined below with my general comments at the end regarding the suitability for publication in Nature Communications.

Point #1) The authors provide a significant amount of new analyses to look at the mechanisms that SMARCA4/2 employ to regulate CCND1 expression. Previously, a weak ChIP association of SMARCA4 with the CCND1 promoter was presented as the mechanism supporting the model. Now ATAC-seq data has been added to show that forced SMARCA4/2 expression can increase chromatin accessibility at the CCND1 promoter and a putative enhancer sequence upstream of the gene has been identified. Additional H3K27Ac data support the ATAC-seq data and the notion that an enhancer sequence is newly opened and active after SMARCA4 re-expression. Interestingly, the putative new enhancer region contains two AP-1 sequences that large scale orthogonal data sets have previously shown to bind c-Jun. The authors show additional data that SMARCA4 may drive JUN expression by associating with the JUN promoter, increasing chromatin accessibility at that promoter, and inducing higher c-Jun protein levels in the cells. The model supported by the data is that SMARCA4/2 expression opens chromatin structure at the CCND1 promoter and an upstream enhancer, while also inducing expression of c-Jun which binds the newly opened enhancer to ultimately support higher CCND1 expression. CCND1 expression levels in turn dictate sensitivity to CDK4/6 inhibitors. These added data do add mechanistic support overall and an interesting model; though the involvement of typical players regulating cell proliferation (i.e. AP-1 transcription factors) diminishes interest.

We appreciate that the reviewer agrees that our new data have provided mechanistic support for our model. We acknowledged that AP-1 transcription factors are known activators of CCND1 in our previous revision. Our new finding regarding this point is that SMARCA4/2 promote CCND1 expression in part by inducing c-Jun.

Point #2) Data to address the the lack of add back of SMARCA4/2 in NCLC cell lines that naturally lack SMARCA4/2 expression during xenografts has been added. The results are supportive of the overall conclusions, although they were only conducted on one cell line. H1299 cells were made into an isogenic pair (Parental line that is SMARCA4 deficient and one that expresses SMARCA4 from a cDNA). Xenografts derived from this cell line are made resistant to palbociclib by SMARCA4 add back supporting the model.

The authors also show that SMARCA4 add back to H1703 cells can confer palbociclib resistance in vitro. The data in figure 6b, d show a very small diminution of H1703 cell growth after SMARCA4 or SMARCA2 add back and the authors claim in their rebuttal that they were too worried that this would impact the ability of the line to form xenografted tumors. However, the relative difference between the parental and SMARCA4 expressing H1299 (Fig 6a) and H1703 (Fig6b) cells seems pretty similar. I just don't see the logic for ruling out one of the lines based on this and for developing an entirely new system for the H1703 cells where they can induce SMARCA4 expression with dox, especially when this strategy failed to help in the end anyway (Reviewer Figure 2).

In contrast to the reviewer's interpretation, forced SMARCA4/2 expression in H1703 did result in clear growth suppression in the colony formation assays (**Figure 6b, d**). We rationalized that this constitutive expression of ectopic SMARCA4 would suppress the establishment of H1703 tumors *in vivo*. Inducible expression system is a well-established method better suited for such situation. This was the reason that we invested extra efforts testing this system.

The authors have ignored my suggestion that they knockdown SMARCA4/2 in cell lines that have high expression and asses the impact on CDK4/6 sensitivity during xenograft analysis.

Knockdown of SMARCA4/2 in cell lines with high expression is toxic with extended long-term culturing (see below), which is not be suitable for *in vivo* drug study. Similar observations have been reported by others (PMID: 22233809). We apologize for not including these negative data in last revised version.

Overall this seems like a halfhearted and haphazard approach to address my comment though the data selected for inclusion in the manuscript does address the concern.

We respectfully disagree with the reviewer. We have carefully performed all suggested experiments except ones with technical infeasibilities discussed above and provided additional supporting data. Our new data have provided new mechanistic insights as requested.

General Comments

In the end, the question remains as to what does this manuscript provide that is additional to the companion manuscript? No doubt, it is clinically important to show that SMARCA4/2 deficient NSCLCs have heightened sensitivity to palbociclib. However, this is a “me too” result if published alongside another paper outlining the initial discovery found using an interesting and somewhat unbiased screening approach. The added chromatin remodeling studies are interesting and likely explain the differential expression of CCND1 and sensitivity to CDK4/6 inhibition. However, some insights are limited and only just beginning to be explored by the authors. For example, new Figure 4 offers only a global overview of the impacts on chromatin accessibility and H3K36Ac after SMARCA4/2 overexpression which would be expected to occur after over-expressing a critical component of chromatin regulating complex that was previously absent. Any detailed insight is not revealed until figure 5. Figures

1,2,3 and 6 are derivatives of the companion paper and establish genotype dependent CDK4/6 sensitivity. So the real novelty here comes solely from figure 5 and the insight is that a canonical transcription factor (AP-1) controls proliferation by increasing expression of a canonical cyclin (CCND1). Weighing the insights gained against the relatively weak approaches that rely largely on human cancer cell lines and overexpression studies, my opinion is that this manuscript falls short of the mark.

We appreciate the reviewer’s recognition of the clinical importance of this study and positive comments for most of the new data provided. However, we would like to clarify below that this study is more than a “me too” result, in addition to the companion manuscript.

First, following the reviewer’s suggestions, we further investigated the mechanisms of cyclin D1 regulation by SMARCA4/2 in this study. We show that SMARCA4/2 regulate cyclin D1 not only by directly remodelling chromatin structure of its gene locus but also by inducing c-Jun. There are all new findings. It is also agreed by the reviewer that our new Figure 5 provide novel mechanism insights.

Second, it is well-known that different cancer types harboring identical oncogenic alterations may respond differently to the same targeted therapeutics, due to the context-dependency (PMID: 22281684; 27648353; 29101114). Furthermore, NSCLC harbors a much more complex mutation landscape compared to genomically quiescent SCCOHT, where SMARCA4 loss is the only recurrent genetic alteration. Therefore, it is not necessarily an expected result that we extended our findings from SCCOHT to NSCLC. Moreover, this extension indicates a good conservation of the synthetic lethality between SMARCA4 loss and CDK4/6 inhibition between cancers. Considering that SMARCA4 is also inactivated in a variety of other cancer types (PMID: 25774356), this study may inspire the exploration of CDK4/6 inhibitors to target these cancers as well.

Finally, unlike SCCOHT where SMARCA2 is always silenced, NSCLC express variable levels of this protein. In the current study, we also investigated and established the role of SMARCA2 in modulating cyclin D1 expression and drug responses to CDK4/6 inhibitors. These new data indicate that SMARCA4/2 loss induce drug sensitivity to CDK4/6 inhibitors through SWI/SNF complexes. Given that other SWI/SNF subunits are frequently inactivated in cancers, our findings in NSCLC may also promote future study of CDK4/6 inhibitors in other SWI/SNF mutant cancers.

Therefore, we believe that this study may have important implications beyond NSCLC and will be of interest to a wide spectrum of clinicians and scientists.

Reviewer #3 (Remarks to the Author):

I have been asked to review the novel set of data (CHIP-seq and ATAC-seq) that have been added during revision to the manuscript. Overall, the data seem of good quality, but my interpretation is limited by several factors. First of all, the authors do not provide any analytical parameters such as number of decuplicated read per sample, alignment rates etc, to judge if the analysis was done appropriately, I could only see heatmaps and IGV snapshot.

We thank the referee for reviewing our new data. We have included the mapping data in the **new Supplementary Table 3** and **4**. The raw and processed reads are also available on GEO (GSE121755) with the following Reviewer Token: qnujmkgw nrwz bop

Secondly, there is something rather odd in figure 4f as there is a stark change in the background in the heatmap, almost as if two heatmaps were generated independently and then fused.

This discontinuity arises from the algorithm used by NGS Plot to organize rows when plotting several datasets side by side. Briefly, Figure 4f was mapped using the ngs.plot.r with default parameters and the -SC global parameter, so that the heat map intensities were equivalent for the three samples. We have included the raw output file “K27AcMetaPlot.heatmap” for the reviewer.

For figure 5 IGV plots, at the bare minimum there should be a RPKM number for each plot, to make sure the comparison is fair.

Scale is now indicated in the upper left in these plots. These are mapped bam-files being plotted (or in the case of the SMARCA4 ChIP, a smoothed bigwig file to match the bigwig file from published H1299 data). For all of the data we generated, the track heights are normalized to read number within each category (e.g. ATAC tracks are normalized to each other, H3K27Ac tracks are normalized to each other) etc. The calculations are shown in the table “Track height normalization” attached to this response letter for the reviewer.

Finally, in figure 4 it would good to have the heatmaps including the common 40k or so sites to have confidence that the acquired ones are similar in terms of enrichment (it would appear so from the CCND1 and JUN IGV snapshots).

We now have included metaplots and heatmaps over these common 40,749 regions in the **new Supplementary Figure 9**. As the reviewer has predicted, ATAC and H3K27Ac enrichment over these regions is fairly similar in the three datasets.

It is also very interesting that almost all the gained ATAC and CHIP signal occur at distal regulatory elements, where the common are much more biased toward promoters. I was wondering if the authors could comment on that.

We have commented on this in Page 9 after citing Figure 4b, noting that enhancer utilization differs more between different cell types than promoter openness, and that SMARCA4 has been observed to regulate enhancer opening in other contexts. Thus, this particular finding is consistent with literature (relevant references are included in the main text).

Table for review: Track height normalization

Sample	Normalization group	Read number	Read number Ratio	Height Figure 5A	Height Figure 5B	Height Figure 5C
ATAC-seq mapping data						
Control H1703	1	17734846	1	140	196	196
SMARCA4-restored H1703	1	38054024	2.145720577	300	420	420
SMARCA2-restored H1703	1	22822582	1.286877935	180	252	252
ChIP-seq mapping data						
Control H1703 H327Ac ChIP	2	26294654	1	35		35
SMARCA4-restored H1703 H3K27Ac ChIP	2	25695976	0.97723195	34		34
SMARCA2-restored H1703 H3K27Ac ChIP	2	24257365	0.922520791	32		32
Control H1703 H3K4me3 ChIP	3	21161530	1	58		58
SMARCA4-restored H1703 H3K4me3 ChIP	3	20592956	0.973131716	56		56
SMARCA2-restored H1703 H3K4me3 ChIP	3	20917582	0.9884721	57		57
SMARCA4-restored H1703 Input for SMARCA4 ChIP	4	5751680	1	2.67		3.22
SMARCA4-restored H1703 SMARCA4 ChIP	4	9919583	1.724640974	4.5		5.56

REVIEWERS' COMMENTS:

Reviewer #3 (Remarks to the Author):

I am satisfied with the author's response

Luca Magnani

REVIEWERS' COMMENTS:

Reviewer #3 (Remarks to the Author):

I am satisfied with the author's response

Luca Magnani

We are pleased to see that this reviewer has responded positively.